

# Yang-Baxter and the Boost: splitting the difference

**Marius de Leeuw⋆, Chiara Paletta†, Anton Pribytok‡, Ana L. Retore▽ and Paul Ryan◇**

School of Mathematics & Hamilton Mathematics Institute
Trinity College Dublin, Dublin, Ireland

⋆ mdeleeuw@maths.tcd.ie, † palettac@maths.tcd.ie, ‡ apribytok@maths.tcd.ie,
▽ retorea@maths.tcd.ie, ◇pryan@maths.tcd.ie

## Abstract

In this paper we continue our classification of regular solutions of the Yang-Baxter equation using the method based on the spin chain boost operator developed in [1]. We provide details on how to find all non-difference form solutions and apply our method to spin chains with local Hilbert space of dimensions two, three and four. We classify all $16 \times 16$ solutions which exhibit $\mathfrak{su}(2) \oplus \mathfrak{su}(2)$ symmetry, which include the one-dimensional Hubbard model and the $S$-matrix of the $AdS_5 \times S^5$ superstring sigma model. In all cases we find interesting novel solutions of the Yang-Baxter equation.

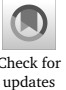
## 1 Introduction

The Yang-Baxter equation (YBE) appears in many different areas of physics [2–5]. It signals the presence of integrability which implies the existence of higher conservation laws. The equation emerges in some form in virtually every area of physics, including condensed matter, statistical physics, (quantum) field theory, string theory and even quantum information theory [6]. The Heisenberg spin chain and the Hubbard model [7] are just some of the famous integrable models and were important for our understanding of low-dimensional statistical and condensed matter systems and, similarly, over the last few years, exceptional progress has been made in understanding the AdS/CFT correspondence [8–10] due to the discovery of integrable structures [11]. Given the clear ubiquity of the Yang-Baxter equation throughout theoretical physics it is clear that understanding and classifying its solutions is a highly interesting and non-trivial task.

The presence of quantum integrability in a given physical model with Hilbert space $\mathbb{C}^n$ is dictated by the existence of a solution $R(u, v) \in \text{End}(\mathbb{C}^n \otimes \mathbb{C}^n)$, dubbed *R-matrix*, of the Yang-Baxter equation, which reads

$$R_{12}(u, v)R_{13}(u, w)R_{23}(v, w) = R_{23}(v, w)R_{13}(u, w)R_{12}(u, v) \tag{1.1}$$

on $\mathbb{C}^n \otimes \mathbb{C}^n \otimes \mathbb{C}^n$ and the subscripts denote which of the three spaces $R$ acts on. The parameters $u, v, w$ are known as spectral parameters with one associated to each of the three spaces. Once $R$ is known one can construct the so-called transfer matrix $t(u, \theta)$ for a spin chain of length $L$ as

$$t(u, \theta) = \text{tr}_a \left( R_{aL}(u, \theta) \ldots R_{a1}(u, \theta) \right), \tag{1.2}$$

which generates an infinite tower of conserved charges ($\mathbb{Q}_i$, $i = 1, \ldots, \infty$) via the expansion

$$\log t(u, \theta) = \mathbb{Q}_1(\theta) + (u - \theta)\mathbb{Q}_2(\theta) + \frac{1}{2}(u - \theta)^2 \mathbb{Q}_3(\theta) + \ldots. \tag{1.3}$$

The parameter $u$ is an auxiliary spectral parameter, whereas the parameter $\theta$ is a physical parameter such as the rapidity of a particle in a scattering process. An immediate property of the YBE is that the charges $\mathbb{Q}_r$ commute:

$$[\mathbb{Q}_r(\theta), \mathbb{Q}_s(\theta)] = 0, \tag{1.4}$$

which is the cornerstone of integrability.

A particularly interesting class of *R*-matrices are the so-called *regular* solutions which are those *R*-matrices $R(u, v)$ which satisfy the *regularity condition* $R_{12}(u, u) = P_{12}$ where $P_{12}$ denotes the permutation operator on the two copies of $\mathbb{C}^n$. The significance of such solutions

is that for the corresponding integrable system, momentum is a conserved charge, or more precisely the tower of conserved charges commutes with the operator of cyclic permutations which is a prevalent feature of many integrable models such as the Hubbard model. In this case the conserved charges $\mathbb{Q}_r(\theta)$ are a sum of densities of range $r$, meaning each density acts on $r$-adjacent spin chain sites. For example, the Hamiltonian $\mathbb{Q}_2(\theta)$ can be written as a sum of nearest-neighbour (range 2) densities $\mathcal{H}_{j,j+1}(\theta)$ as

$$\mathbb{Q}_2(\theta) = \sum_{j=1}^{L} \mathcal{H}_{j,j+1}(\theta) \tag{1.5}$$

and periodic boundary conditions are imposed, that is $\mathcal{H}_{L,L+1}(\theta) = \mathcal{H}_{L,1}(\theta)$. This density is itself related to the $R$-matrix in a very simple way:

$$\mathcal{H}_{12}(\theta) = P_{12}\partial_u R_{12}(u,\theta)|_{u\to\theta}. \tag{1.6}$$

Hence, the moment one knows the $R$-matrix one knows the Hamiltonian and the dynamics of the system.

Throughout the history of quantum integrable systems numerous different approaches have been developed for finding solutions of the Yang-Baxter equation. In the early days a very fruitful approach has been through requiring the solutions to have certain symmetries [12–14]. For example, if we wish for the Hamiltonian $\mathbb{Q}_2$ to commute with the generators $\mathfrak{a}$ of some Lie algebra $\mathfrak{g}$ then one should impose that $[R(u,v), \mathfrak{a} \otimes 1 + 1 \otimes \mathfrak{a}] = 0$. More generally given some bialgebra $\mathcal{A}$ we require that $\Delta^{\text{op}}(\mathfrak{a})R(u,v) = R(u,v)\Delta(\mathfrak{a})$ where $\Delta$ and $\Delta^{\text{op}}$ denote the coproduct and opposite coproduct related by conjugation on $\mathcal{A}$, respectively. In many cases this is enough to completely fix $R$ up to a small number of functions, drastically simplifying the construction, as was demonstrated in the case of AdS/CFT integrable systems [15–22], see also [23] for recent developments using this approach. Of course, this approach first requires one to know what the corresponding symmetry is and there are $R$-matrices which may have no such symmetry at all. Still within the realm of algebra, a more abstract approach is that of Baxterisation which initially appeared in the realm of knot theory [24–27] and consists of constructing solutions of the YBE as representations of certain algebras, for example Hecke algebras and Temperly-Lieb algebras. Numerous different $R$-matrices have been obtained in this way [28–34] and further advancements have also been achieved recently [35–37].

A more hands-on approach is to simply try and solve the Yang-Baxter equation directly. The upside to this is that in principle one can obtain all solutions in this way, but this is contrasted with the enormous difficulty of solving cubic functional equations. This approach is usually supplemented with differentiating[1] the YBE and reducing the cubic functional equations to a system of coupled partial differential equations. This approach has recently been used to provide a full classification of $R$-matrix of size $4 \times 4$ so-called 8-and-lower-vertex models [38] obeying the difference property $R(u,v) = R(u-v)$ and to obtain certain $9 \times 9$ models [39] whose $R$-matrix satisfies the so-called *ice rule* but it quickly becomes unwieldy as the size of the $R$-matrix increases.

In this paper we follow a bottom-up approach which we have developed in a series of recent papers [1, 40, 41] and further develop here. In our approach, instead of starting with the $R$-matrix and using it to find the Hamiltonian and the corresponding dynamics, we start with the Hamiltonian and use it to obtain the $R$-matrix. The mechanism for carrying out this procedure hinges on the so-called *boost automorphism* [42–44] which is an alternative, yet equivalent,

---

[1]Assuming differentiability of the $R$-matrix in a neighbourhood of some point is actually not a loss of generality, since this must be the case in order to obtain the conserved charges from the transfer matrix in a power series expansion. Of course there can be $R$-matrices which are not differentiable but to our knowledge these do not have a physical interpretation.

way to generate the tower of conserved charges for regular integrable models without the need to construct the transfer matrix and expand it. The boost automorphism $\mathcal{B}[\mathbb{Q}_2]$, or simply the boost operator, is defined by

$$\mathcal{B}[\mathbb{Q}_2] := \partial_\theta + \sum_{n=-\infty}^{\infty} n\mathcal{H}_{n,n+1}(\theta). \tag{1.7}$$

The infinite sum should be interpreted in a formal sense but what we are interested is not the boost operator itself but rather its commutator with the tower of conserved charges which is perfectly well-defined even for finite chains. In fact, it can be shown that, see Appendix A

$$\mathbb{Q}_{r+1} = [\mathcal{B}[\mathbb{Q}_2], \mathbb{Q}_r], \quad r > 1. \tag{1.8}$$

Hence, by knowing just the Hamiltonian density $\mathcal{H}_{12}(\theta)$ we can construct the full tower of commuting conserved charges directly and our approach is based on exploiting this observation. Namely, instead of starting with a solution of the YBE we will start with a generic operator on $\mathbb{C}^n \otimes \mathbb{C}^n$ which we identify as a Hamiltonian density $\mathcal{H}_{12}(\theta)$ and construct the corresponding global charge $\mathbb{Q}_2(\theta)$. We will then use the boost operator to construct $\mathbb{Q}_3(\theta)$ by the relation (1.8). A priori there is no reason for the two constructed operators $\mathbb{Q}_2$ and $\mathbb{Q}_3$ to commute with each other, but if we impose this it will place a number of constraints on the entries of the density $\mathcal{H}_{12}$ in the form of a system of ODEs. We then solve the set of constraints and show that the resulting Hamiltonian defines an integrable system, meaning it can be obtained from a solution of the YBE. In order to do this we use the so-called *Sutherland equations* which are obtained from the YBE and read

$$[R_{13}R_{23}, \mathcal{H}_{12}(u)] = \dot{R}_{13}R_{23} - R_{13}\dot{R}_{23}, \tag{1.9}$$

$$[R_{13}R_{12}, \mathcal{H}_{23}(v)] = R_{13}R'_{12} - R'_{13}R_{12}, \tag{1.10}$$

where in each of the above Sutherland equations $R_{ij} := R_{ij}(u, v)$ and $\dot{R}$ and $R'$ denote the derivatives of $R$ with respect to the first and second variable respectively. The Sutherland equations constitute two sets of ODEs for the entries of the $R$-matrix and the boundary conditions are fixed by the requirement of regularity $R(u, u) = P$ and the fact that the Hamiltonian density can be obtained from the $R$-matrix by the expansion

$$R_{12}(u, v) = P_{12}\left(1 + (u - v)\mathcal{H}_{12}\left(\frac{u + v}{2}\right) + \mathcal{O}((u - v)^2)\right). \tag{1.11}$$

Hence, in effect, solving the condition $[\mathbb{Q}_2(\theta), \mathbb{Q}_3(\theta)] = 0$ singles initial conditions for the Sutherland equations which have a chance to be consistent with the existence of an $R$-matrix. What is remarkable is that *all* initial conditions obtained in this way lead to an $R$-matrix, at least for the cases discussed in this paper and in [1, 40, 41].

Let us remark that our approach is actually not the first which uses the Hamiltonian as a starting point for constructing integrable systems and $R$-matrices. In a series of papers [45–47] the authors present a method for determining if a given nearest-neighbour Hamiltonian system is solvable by the coordinate Bethe ansatz [48, 49] and also produce $R$-matrices for some of these systems. There is also the earlier work [50] where an iterative procedure for reconstructing the $R$-matrix from the Hamiltonian was developed for models where the $R$-matrix satisfies the difference property $R(u, v) = R(u - v)$, as well as the work [51, 52]. In the case of 1+1-dimensional integrable field theories such $R$-matrices correspond to $S$-matrices which are Poincaré invariant and include integrable systems such as the XYZ spin chain and its derivatives and Zamolodchikov's $O(N)$ sigma model. When one restricts to this case, our procedure

described above for constructing $R$-matrices from Hamiltonians simplifies enormously. In particular the conserved charges $\mathbb{Q}_r(\theta)$ become independent of $\theta$ and so the set of ODEs arising from the condition $[\mathbb{Q}_2(\theta), \mathbb{Q}_3(\theta)] = 0$ reduces to a set of coupled cubic polynomial equations. This simplification was exploited in the papers [1, 40] in order to find a plethora of new integrable systems with a range of interesting physical properties. In this paper, in order to demonstrate the full power of our approach we will not impose the difference property and consider the most general $R$-matrices. One of the main results of this paper is that the single consistency condition $[\mathbb{Q}_2(\theta), \mathbb{Q}_3(\theta)] = 0$ on the Hamiltonian is enough in order to completely determine the $R$-matrix even in the absence of the difference property, which goes back to a conjecture of [53]. The analysis of the most general possible $R$-matrices using the boost approach was initiated in [41] and here we continue that analysis. For the higher-rank case however, that is beyond $4 \times 4$ $R$-matrices, we will impose that our $R$-matrices have certain symmetries in order to render the calculations tractable. For $9 \times 9$ $R$-matrices we will impose that our $R$-matrices commute with the Cartan subalgebra of $\mathfrak{su}(3)$ and for $16 \times 16$ $R$-matrices we will impose $\mathfrak{su}(2) \oplus \mathfrak{su}(2)$ symmetry which appears in various interesting models such as the $\mathfrak{su}(4)$ Heisenberg XXX spin chain [13], the Hubbard model [54] and the AdS/CFT $S$-matrix [15] and the related Shastry $R$-matrix [55]. Furthermore, for $16 \times 16$ models we determine all possible integrable models which preserve fermion number and a Hamiltonian based on a generalisation of the usual Hubbard model. Let us point out that such restrictions are not strictly necessary to implement our approach, but due to the fact that our method produces huge numbers of integrable systems, many of which are trivially related as will be explained in the main text, providing a full classification of all possible $9 \times 9$ and $16 \times 16$ $R$-matrices is highly difficult and so we limit ourselves to a subset of models which are physically interesting.

This paper is organised as follows. In Section (2) we review the procedure developed in [1, 40, 41] and outlined here in the introduction and explain in detail how to go from a generic Hamiltonian to an integrable Hamiltonian and subsequently find a solution of the Yang-Baxter equation. In Section (3) we will apply our procedure to models with two-dimensional local Hilbert space which produces $R$-matrices of size $4 \times 4$ which were initially presented in our letter [41]. In Sections (4) and (5) we discuss models with three- and four-dimensional local Hilbert spaces with certain symmetries imposed, namely $\mathfrak{u}(1) \oplus \mathfrak{u}(1) \oplus \mathfrak{u}(1) \subset \mathfrak{su}(3)$ and $\mathfrak{su}(2) \oplus \mathfrak{su}(2)$ respectively as well as the generalised Hubbard models mentioned earlier. Finally, we discuss some further directions for research. In Appendix A we review the construction of the charges $\mathbb{Q}_r(\theta)$ using the boost operator.

We have attached a `Mathematica` notebook to the arxiv submission of this paper which contains all of the $R$-matrices obtained in this paper as well as those in [1, 40, 41] together with the corresponding Hamiltonians and commands to check various properties. In Appendix B we provide some details on this notebook.

## 2 Set-up and method

In this section we will give more details on our method and discuss an explicit example to illustrate the procedure that we follow.

### 2.1 Method

As described in the introduction our starting point is a nearest-neighbour Hamiltonian density $\mathcal{H}_{12}(\theta)$ on $\mathbb{C}^n \otimes \mathbb{C}^n$. Using the density we construct the full Hamiltonian $\mathbb{H}(\theta) = \mathbb{Q}_2(\theta)$ on a

spin chain of length 4

$$\mathbb{Q}_2(\theta) = \sum_{j=1}^{4} \mathcal{H}_{j,j+1}(\theta) \tag{2.1}$$

and we identify sites $4 + j := j$, $j = 1, \ldots, 4$. The reason for our restriction to length 4 will be explained below. From now on, for shortness, we will sometimes omit the $\theta$ dependence on the density Hamiltonian.

Using the boost operator (1.7) we construct $\mathbb{Q}_3(\theta)$, which in this case is given explicitly by

$$\mathbb{Q}_3(\theta) = -\sum_{j=1}^{4} [\mathcal{H}_{j,j+1}(\theta), \mathcal{H}_{j+1,j+2}(\theta)] + \partial_\theta \mathbb{Q}_2(\theta). \tag{2.2}$$

As was described in the introduction $\mathbb{Q}_3(\theta)$ can be written as a sum of range 3 densities $\mathcal{Q}_{j,j+1,j+2}(\theta)$:

$$\mathbb{Q}_3(\theta) = \sum_{j=1}^{4} \mathcal{Q}_{j,j+1,j+2}(\theta). \tag{2.3}$$

Next, we impose the condition $[\mathbb{Q}_2(\theta), \mathbb{Q}_3(\theta)] = 0$ which is a necessary condition for the model to be integrable and solve the resulting set of ODEs for the entries of $\mathcal{H}_{12}$. Then we plug the obtained density $\mathcal{H}_{12}(\theta)$ into the Sutherland equations described in the introduction

$$[R_{13}R_{23}, \mathcal{H}_{12}(u)] = \dot{R}_{13}R_{23} - R_{13}\dot{R}_{23}, \tag{2.4}$$

$$[R_{13}R_{12}, \mathcal{H}_{23}(v)] = R_{13}R'_{12} - R'_{13}R_{12}, \tag{2.5}$$

solve the resulting set of differential equations for $R(u, v)$ and fix this uniquely by using the boundary conditions

$$R(u, u) = P, \quad \dot{R}(u, v)|_{v \to u} = P\mathcal{H}(u). \tag{2.6}$$

Finally, we check that the Yang-Baxter equation is indeed satisfied. This procedure can be conveniently outlined in the diagram of Figure 1.

At this point we would also like to highlight that in case the $R$-matrix is of difference form (that is satisfy the difference form property), the Hamiltonian will not depend on the spectral parameter and hence all derivative terms drop out. This means that the integrability condition simply reduces to a set of polynomial equations and that the Sutherland reduces to ordinary (non-linear) differential equations since $R$ effectively only depends on one variable.

Finally a small comment is due on why we restrict to spin chains of length 4. Since $\mathbb{Q}_2$ is a sum of range 2 densities and $\mathbb{Q}_3$ is a sum of range 3 densities the non-vanishing terms in their commutator $[\mathbb{Q}_2, \mathbb{Q}_3]$ is a sum of densities of range $2 + 3 - 1 = 4$. Hence, if we restrict to a spin chain of length 3 say, then these non-zero commutators will effectively wrap around the spin chain producing cancellations which do not happen in general. Hence for our construction we must consider spin chains of at least length 4 in order to avoid this happening. Alternatively, we can also derive the same system of equations by simply looking at the densities. In case one needs to consider the commutation relations between higher conserved charges, the length of the spin chain needs to be adjusted accordingly - if one wants to consider the commutator $[\mathbb{Q}_n, \mathbb{Q}_m]$ then a spin chain of length $L = n + m - 1$ should be considered.

## 2.2 Identifications

As we explained in [1, 40, 41], our approach of solving the YBE by imposing the condition $[\mathbb{Q}_2, \mathbb{Q}_3] = 0$ leads to quite a large redundancy in solutions. Specifically, for a given integrable Hamiltonian density (or $R$-matrix) there are various transformations one can do which preserve

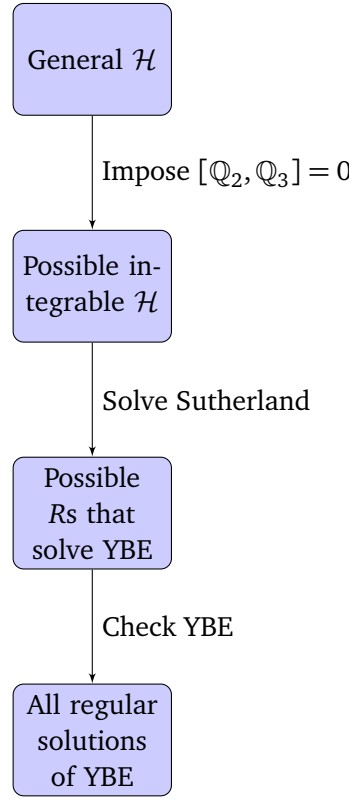

Figure 1: Flowchart of determining regular solutions of the Yang-Baxter equation.

this condition and preserve regularity. Hence in what follows we will only concern ourselves with a single representative of equivalence classes of Hamiltonians and $R$-matrices. We now describe the transformations which lead to equivalent solutions.

**Local basis transformation**    If $R(u, v)$ is a solution of the Yang-Baxter equation and $V$ an invertible matrix depending on one spectral parameter and with the same size of the $R$-matrix, then we can generate another solution by defining

$$R^{(V)}(u, v) = \left[ V(u) \otimes V(v) \right] R(u, v) \left[ V(u) \otimes V(v) \right]^{-1}. \tag{2.7}$$

This new solution is trivially compatible with regularity and just corresponds to a change of basis on each site. On the level of the Hamiltonian it gives rise to a new integrable Hamiltonian which takes the form

$$\mathcal{H}^{(V)} = \left[ V \otimes V \right] \mathcal{H} \left[ V \otimes V \right]^{-1} - \left[ \dot{V} V^{-1} \otimes I - I \otimes \dot{V} V^{-1} \right], \tag{2.8}$$

where everything is evaluated at $\theta$ and $I$ is the identity matrix. In particular, we see that terms of the form $A \otimes I - I \otimes A$ in the Hamiltonian can be removed by performing the basis transformation (2.8) with the matrix $V(u)$ satisfying $\dot{V} = AV$ which can be solved by means of a path-ordered exponential.

**Reparameterization**    If $R(u, v)$ is a solution, then $R(g(u), g(v))$ clearly is a solution of the YBE as well. This transformation affects the normalization of the Hamiltonian since by the chain rule the logarithmic derivative of $R$ will give an extra factor $\dot{g}$, so that

$$\mathcal{H}(u) \mapsto \dot{g} \mathcal{H}(g(u)). \tag{2.9}$$

Notice furthermore that this will similarly affect the derivative term in the boost operator. We are also free to reparameterize any other functions and constants in both the $R$-matrix and Hamiltonian. For instance the $R$-matrices from [56,57] can be obtained by using a reparameterization of the usual XXX $R$-matrix.

**Normalization**   We can normalize the $R$-matrix in any way we want since multiplying any solution $R$ of the YBE by an arbitrary function $g$ is clearly allowed. On the level of the Hamiltonian this corresponds to a simple shift of the Hamiltonian

$$\mathcal{H} \mapsto \mathcal{H} + \dot{g} I, \tag{2.10}$$

where $I$ is the identity matrix. We have imposed $g(\theta, \theta) = 1$ in order to preserve $R(\theta, \theta) = P$.

**Discrete transformations**   It is straightforward to see that for any solution $R(u, v)$ of the Yang-Baxter equation, $PR(u, v)P, R^T(u, v)$ and $PR^T(u, v)P$ are solutions as well.

All the above transformations are universal and hold for any integrable model. Moreover, they have a trivial effect on the spectrum, which means that they basically describe the same physical model. Additionally, there are some transformations called twists that we can use for identifications that are model dependent. Twists generically change the spectrum and more generally the physical properties of the integrable model in a non-trivial way. However, on the level of the $R$-matrix a twist is a simple transformation.

**Twists**   If $U(u)$ is an invertible $n \times n$ matrix which satisfies $[U(u) \otimes U(v), R_{12}(u, v)] = 0$ then it can be shown that

$$U_2(u)R_{12}(u, v)U_1(v)^{-1} \tag{2.11}$$

is a solution of the YBE provided $R$ is. Note that much more general transformations which preserve the YBE can be obtained by combining (2.11) together with other transformations. For example, if both $U$ and $V$ are constant invertible matrices satisfying $[U \otimes U, R_{12}] = 0 = [V \otimes V, R_{12}]$ then the following is also a solution

$$U_1 V_2 R_{12} U_2^{-1} V_1^{-1}, \tag{2.12}$$

which can be obtained by applying (2.11) together with a similarity transformation and applying (2.11) again. We will refer to any transformation obtained by combining (2.11) with the other transformations mentioned above as a *twist*.

Under the transformation (2.11) the Hamiltonian density $\mathcal{H}_{12}$ transforms as

$$\mathcal{H}_{12} \mapsto U_1 \mathcal{H}_{12} U_1^{-1} + \dot{U}_1 U_1^{-1} \tag{2.13}$$

and the analogue of the condition $[U(u) \otimes U(v), R_{12}(u, v)] = 0$ for the Hamiltonian density can be easily worked out to be

$$[U_1 U_2, \mathcal{H}_{12}] = \dot{U}_1 U_2 - U_1 \dot{U}_2. \tag{2.14}$$

Alternatively this relation may be derived by plugging the twisted $R$-matrix (2.11) and Hamiltonian (2.13) into the Sutherland equations (1.9) and sending $v \to u$, which is not surprising given the striking similarity between (2.14) and the Sutherland equations.

Finally, there can be other, model dependent, twists such as Drinfeld twists [58,59]. Moreover, the condition $[U(u) \otimes U(v), R_{12}(u, v)] = 0$ can also be extended to, for instance, depend on two twists $U, V$ or the spectral dependence of the twist can be modified. We will usually only use standard twists (2.11) unless stated otherwise.

## 2.3 Example

As a demonstration of our method let us work out an example in full detail. From here on we will use the following notation:

- $h_i(u)$ are matrix elements of $\mathcal{H}(u)$

- $\dot{h}_i(u) = \partial_u h_i(u)$

- $H_i(u) = \int_0^u h_i$ and $H_i(u,v) = \int_v^u h_i = H_i(u) - H_i(v)$

- $r_i(u,v)$ are matrix elements of $R(u,v)$

- $\dot{r}_i(u,v) = \partial_u r_i(u,v)$ and $r'_i(u,v) = \partial_v r_i(u,v)$.

**Hamiltonian** Let us classify all regular solutions of the YBE whose Hamiltonian densities have the following form

$$\mathcal{H}_{12}(\theta) = \begin{pmatrix} 0 & 0 & 0 & 0 \\ 0 & h_1(\theta) & h_3(\theta) & 0 \\ 0 & h_4(\theta) & h_2(\theta) & 0 \\ 0 & 0 & 0 & 0 \end{pmatrix}. \tag{2.15}$$

From the boost operator construction we find that the corresponding charge $\mathbb{Q}_3$ has density

$$\mathcal{Q}_{123}(\theta) = \begin{pmatrix} 0 & 0 & 0 & 0 & 0 & 0 & 0 & 0 \\ 0 & 0 & -h_1 h_3 & 0 & -h_3^2 & 0 & 0 & 0 \\ 0 & h_1 h_4 & \dot{h}_1 & 0 & \dot{h}_3 - h_2 h_3 & 0 & 0 & 0 \\ 0 & 0 & 0 & \dot{h}_1 & 0 & \dot{h}_3 + h_1 h_3 & h_3^2 & 0 \\ 0 & h_4^2 & \dot{h}_4 + h_2 h_4 & 0 & \dot{h}_2 & 0 & 0 & 0 \\ 0 & 0 & 0 & \dot{h}_4 - h_1 h_4 & 0 & \dot{h}_2 & h_2 h_3 & 0 \\ 0 & 0 & 0 & -h_4^2 & 0 & -h_2 h_4 & 0 & 0 \\ 0 & 0 & 0 & 0 & 0 & 0 & 0 & 0 \end{pmatrix} \tag{2.16}$$

and is quadratic in the components $h_i(\theta)$ of the Hamiltonian density $\mathcal{H}$. We have suppressed the $\theta$ dependence.

The next step is to impose $[\mathbb{Q}_2(\theta), \mathbb{Q}_3(\theta)] = 0$ which gives the equations

$$\dot{h}_3(h_1 + h_2) = (\dot{h}_1 + \dot{h}_2)h_3, \qquad \dot{h}_4(h_1 + h_2) = (\dot{h}_1 + \dot{h}_2)h_4. \tag{2.17}$$

These are solved by

$$h_3 = \frac{c_3}{2}(h_1 + h_2), \qquad h_4 = \frac{c_4}{2}(h_1 + h_2), \tag{2.18}$$

for some constants $c_{3,4}$. Thus we find that if $\mathcal{H}_{12}$ is to be obtained from an $R$-matrix it must have the form

$$\mathcal{H}(\theta) = \begin{pmatrix} 0 & 0 & 0 & 0 \\ 0 & h_1 & \frac{c_3}{2}(h_1 + h_2) & 0 \\ 0 & \frac{c_4}{2}(h_1 + h_2) & h_2 & 0 \\ 0 & 0 & 0 & 0 \end{pmatrix}. \tag{2.19}$$

**R-matrix** We make an ansatz for our R-matrix of the following form

$$R = \begin{pmatrix} r_1 & 0 & 0 & 0 \\ 0 & r_2 & r_3 & 0 \\ 0 & r_4 & r_5 & 0 \\ 0 & 0 & 0 & r_6 \end{pmatrix}. \tag{2.20}$$

We will first solve the Sutherland equations using brute force before using identifications to greatly simplify the process. The Sutherland equations (1.9) give the following independent set of PDEs

$$c_3 r_2 r_6 = c_4 r_1 r_5, \qquad \frac{\dot{r}_2}{r_2} = \frac{\dot{r}_3}{r_3} + h_1 + h_3 \frac{r_6}{r_5}, \qquad \frac{\dot{r}_4}{r_4} = \frac{\dot{r}_3}{r_3} + h_1 - h_2, \qquad \frac{\dot{r}_1}{r_1} = \frac{\dot{r}_6}{r_6}, \tag{2.21}$$

$$\frac{\dot{r}_2}{r_2} = \frac{\dot{r}_5}{r_5}, \qquad \frac{\dot{r}_3}{r_3} = \frac{\dot{r}_1}{r_1} + h_2 + h_3 \frac{r_2}{r_1}, \qquad \frac{c_3}{2} \left[ \frac{r_4 r_3}{r_1 r_5} - \frac{r_6}{r_5} - \frac{r_2}{r_1} \right] = 1. \tag{2.22}$$

From this we see that

$$r_6 = A r_1, \qquad r_5 = B r_2 \qquad \Rightarrow \qquad A c_3 = B c_4. \tag{2.23}$$

Since we need to impose regularity $R(u, u) = P$, we find that $A = 1$ and $B = c_3/c_4$. Next, we derive that

$$r_4 = r_3 e^{H_1(u,v) - H_2(u,v)} \qquad \text{with} \qquad H_i(u, v) = \int_v^u h_i. \tag{2.24}$$

We are then left with three unsolved PDEs

$$\frac{\dot{r}_2}{r_2} = \frac{\dot{r}_3}{r_3} + h_1 + h_3 \frac{r_6}{r_5}, \qquad \frac{\dot{r}_3}{r_3} = \frac{\dot{r}_1}{r_1} + h_2 + h_3 \frac{r_2}{r_1}, \qquad \frac{c_3}{2} \left[ \frac{r_4 r_3}{r_1 r_5} - \frac{r_1}{r_5} - \frac{r_2}{r_1} \right] = 1. \tag{2.25}$$

In order to solve these we redefine

$$r_1 \mapsto r_3 \left( \tilde{r}_1 - \frac{\tilde{r}_2}{c_4} \right), \qquad r_2 \mapsto r_3 \tilde{r}_2, \qquad r_3 \mapsto r_3 \tag{2.26}$$

so that the last equation becomes

$$c_4^2 e^{H_1 - H_2} = c_4^2 \tilde{r}_1^2 + \omega^2 \tilde{r}_2^2, \tag{2.27}$$

where $H_i = H_i(u) - H_i(v)$ and we have put $\omega^2 = c_3 c_4 - 1$. This equation can now be most conveniently solved by substituting cylindrical coordinates, so that we find

$$\tilde{r}_1 = e^{\frac{H_1 - H_2}{2}} \cos \phi, \qquad \tilde{r}_2 = e^{\frac{H_1 - H_2}{2}} \frac{c_4}{\omega} \sin \phi, \tag{2.28}$$

for some function $\phi$ to be determined by the remaining two differential equations. Notice that this is an overdetermined system. Plugging (2.28) then back into the remaining Sutherland equations gives the following

$$\frac{\dot{\phi}}{\omega} = \frac{h_1 + h_2}{2}, \tag{2.29}$$

which is easily solved upon using the boundary condition that $\phi(u, u) = 0$.

Setting $H_\pm(u, v) = \frac{H_1(u,v) \pm H_2(u,v)}{2}$ and combining everything we are left with the following R-matrix

$$R = e^{H_+} \begin{pmatrix} \cos \omega H_+ - \frac{\sin \omega H_+}{\omega} & 0 & 0 & \\ 0 & c_4 \frac{\sin \omega H_+}{\omega} & e^{-H_-} & 0 \\ 0 & e^{H_-} & c_3 \frac{\sin \omega H_+}{\omega} & 0 \\ 0 & 0 & 0 & \cos \omega H_+ - \frac{\sin \omega H_+}{\omega} \end{pmatrix} \tag{2.30}$$

after choosing the overall normalisation $r_3$ to correctly reproduce the Hamiltonian. Owing to the dependence on both $H_+$ and $H_-$, this $R$-matrix is manifestly of non-difference form. It is straightforward to check that $R$ indeed satisfies the Yang-Baxter equation and that its logarithmic derivative gives the density Hamiltonian (2.19).

**Using identifications**   The above method of finding the $R$-matrix can be greatly simplified if we use some identifications that relate various solutions of the Yang-Baxter equation that we discussed in the previous section.

We start from (2.19) and use a local basis transformation to set $h_1 = h_2$. This is achieved using the matrix $V(\theta)$ with

$$V(\theta) = \exp\left(\frac{1}{2}H_-(\theta)\sigma_z\right), \quad H_\pm(\theta) = \frac{1}{2}(H_1(\theta) \pm H_2(\theta)) \tag{2.31}$$

together with the transformation law (2.8).

Next, we use reparameterization symmetry to set $h_1 = h_2 = 1$. Thus, it follows that all the entries of the Hamiltonian are constant and the resulting Hamiltonian density has the form

$$\mathcal{H}(\theta) = \begin{pmatrix} 0 & 0 & 0 & 0 \\ 0 & 1 & c_3 & 0 \\ 0 & c_4 & 1 & 0 \\ 0 & 0 & 0 & 0 \end{pmatrix}. \tag{2.32}$$

Moreover, we can use a twist and set $c_3 = c_4 = c$. Indeed, it is trivial to check that the twist condition (2.14) is satisfied for any constant invertible diagonal matrix $U$ and the matrix

$$U = \text{diag}\left(\sqrt{c_4}, \sqrt{c_3}\right), \tag{2.33}$$

can be used to bring the Hamiltonian density to the form

$$\mathcal{H}(\theta) = \begin{pmatrix} 0 & 0 & 0 & 0 \\ 0 & 1 & c & 0 \\ 0 & c & 1 & 0 \\ 0 & 0 & 0 & 0 \end{pmatrix}, \tag{2.34}$$

after applying $\mathcal{H}_{12} \mapsto U_1 \mathcal{H}_{12} U_1^{-1}$.

The Sutherland equations are now also easily solved since all the coefficients of the Hamiltonian are simply constants. As a consequence, the $R$-matrix is of difference form and is given by the usual XXZ solution. Putting $\omega^2 = c^2 - 1$ we find

$$R = e^u \begin{pmatrix} \cos \omega u - \frac{\sin \omega u}{\omega} & 0 & 0 & \\ 0 & c\frac{\sin \omega u}{\omega} & 1 & 0 \\ 0 & 1 & c\frac{\sin \omega u}{\omega} & 0 \\ 0 & 0 & 0 & \cos \omega u - \frac{\sin \omega u}{\omega} \end{pmatrix}. \tag{2.35}$$

In order to see that this solution is equivalent to the solution (2.30), let us undo the identifications that we performed to make the Hamiltonian constant. First we undo the twist and apply $R_{12} \mapsto U_2^{-1} R_{12} U_1$ to (2.35) and put $c = \sqrt{c_3}\sqrt{c_4}$ so that we arrive at the $R$-matrix for the Hamiltonian (2.32). Next we reparameterize

$$u \mapsto H_+(u) \tag{2.36}$$

and finally we apply the inverse of the local basis transformation (2.31), immediately obtaining (2.30).

**Difference vs. Non-difference**   After using all the identifications, we see that (2.30) is actually just an $R$-matrix of difference form in disguise. The non-difference nature of the rapidity dependence of the $R$-matrix only resides in local basis transformations, a rescaling and a reparameterization. These can obviously be applied to any solution of difference form to generate a non-difference form solution. In the remainder of this work we will also encounter models which are genuinely of non-difference form, but it is easy to see already at the level of the Hamiltonian if this is the case. More precisely, after solving the integrability condition $[\mathbb{Q}_2(\theta), \mathbb{Q}_3(\theta)] = 0$ our Hamiltonian will depend on a number of free functions. One will usually correspond to a shift, one can be absorbed in a reparameterization of the spectral parameter and then remains a number that can be absorbed by local basis transformations and potentially twists. The exact number of the latter will depend on the set-up. Thus in case of (2.19), we count 2 free functions $h_1, h_2$ and we could have already at that point concluded that the underlying model was actually of difference form.

# 3 Two-dimensional local Hilbert space

We will now apply our approach to the classification of various different integrable systems. The first case we will consider will be the case where the local Hilbert space has dimension two and, consequently, the $R$-matrix is of size $4 \times 4$. In [41] we classified all solutions of the Yang-Baxter equation of 8-and-lower-vertex type and we review this case now for completeness. Further details can be found in [41].

## 3.1 8-and-lower-vertex models

8-and-lower-vertex type models are solutions of the form

$$R = \begin{pmatrix} r_1 & 0 & 0 & r_8 \\ 0 & r_2 & r_6 & 0 \\ 0 & r_5 & r_3 & 0 \\ r_7 & 0 & 0 & r_4 \end{pmatrix}, \tag{3.1}$$

and, consequently, the corresponding Hamiltonian densities are of the form

$$\begin{aligned}
\mathcal{H} =\ & h_1 \, \mathbb{1} + h_2 (\sigma_z \otimes \mathbb{1} - \mathbb{1} \otimes \sigma_z) + h_3 \sigma_+ \otimes \sigma_- + h_4 \sigma_- \otimes \sigma_+ \\
& + h_5 (\sigma_z \otimes \mathbb{1} + \mathbb{1} \otimes \sigma_z) + h_6 \sigma_z \otimes \sigma_z + h_7 \sigma_- \otimes \sigma_- + h_8 \sigma_+ \otimes \sigma_+.
\end{aligned} \tag{3.2}$$

We will briefly recap the solutions of the Yang-Baxter equation of this type that we found. After identifying solutions, we found only four different types of integrable $4 \times 4$ Hamiltonians that solve the integrability condition $[\mathbb{Q}_2(\theta), \mathbb{Q}_3(\theta)] = 0$:

- 6-vertex A, $h_6 \neq 0$ and $h_7 = h_8 = 0$

- 6-vertex B, $h_6 = h_7 = h_8 = 0$

- 8-vertex A, $h_6 \neq 0, h_7 \neq 0, h_8 \neq 0$

- 8-vertex B, $h_6 = 0$ and $h_7 \neq 0, h_8 \neq 0$.

Let us discuss the models in more detail.

**6-vertex A**   Setting $h_7 = h_8 = 0$ and assuming $h_6 \neq 0$ we find that $[\mathbb{Q}_2(\theta), \mathbb{Q}_3(\theta)] = 0$ is satisfied if and only if

$$h_3 = c_3 h_6 e^{4H_5}, \qquad\qquad h_4 = c_4 h_6 e^{-4H_5}, \qquad\qquad (3.3)$$

where $c_{3,4}$ are constants. The Hamiltonian is actually equivalent to that of the XXZ spin chain. Indeed, by applying a local basis transformation, twist, reparameterization and normalization we can bring the Hamiltonian density to the form

$$\mathcal{H} = \begin{pmatrix} 0 & 0 & 0 & 0 \\ 0 & 1 & c & 0 \\ 0 & c & 1 & 0 \\ 0 & 0 & 0 & 0 \end{pmatrix}, \qquad\qquad (3.4)$$

which is precisely the Hamiltonian density (2.34), and so its $R$-matrix is given by (2.35). Notice that this solution also contains the most general diagonal Hamiltonian since only the off-diagonal elements $h_{3,4}$ are restricted by the integrability condition.

**6-vertex B**   If we take $h_6 = h_7 = h_8 = 0$ then it makes the Hamiltonian satisfy $[\mathbb{Q}_2, \mathbb{Q}_3] = 0$ for *any* choice of $h_1, \dots, h_5$. So, the Hamiltonian depends on five free functions. Three of these functions can be absorbed in identifications. In particular, a local basis transformation ($h_2$), a normalization ($h_1$) and a reparameterization of the spectral parameter ($h_3$). Moreover, it is convenient to redefine $h_5 \to \frac{1}{2} h_4 h_5$.

We normalize the $R$-matrix such that $r_5 = 1$ and then it follows from the Sutherland equations (1.9) that

$$r_7 = r_8 = 0, \quad r_6 = 1, \quad \dot{r}_2 = h_4(r_1 - h_5 r_2), \quad \dot{r}_4 = -h_4(r_3 + h_5 r_4), \quad r_1 r_4 + r_2 r_3 = 1, \quad (3.5)$$

while $r_4$ satisfies the second order version of the Riccati equation

$$\ddot{r}_4 - \frac{\dot{h}_4}{h_4} \dot{r}_4 + h_4 r_4 \left[ h_3 + \dot{h}_5 - h_4 h_5^2 \right] = 0. \qquad\qquad (3.6)$$

We now introduce a reparameterization of the spectral parameter

$$u_i \mapsto x_i = \int^{u_i} \frac{\dot{h}_5}{h_4 h_5^2 - h_3}, \qquad\qquad (3.7)$$

which kills the non-derivative term in the Riccati equation and removes the explicit dependence on $h_3$. It is then straightforward to solve our system of differential equations to find

$$r_2(x, y) = H_4(x, y), \qquad\qquad (3.8)$$
$$r_1(x, y) = 1 + h_5(x) H_4(x, y), \qquad\qquad (3.9)$$
$$r_3(x, y) = h_5(x) h_5(y) H_4(x, y) - h_5(x) + h_5(y), \qquad\qquad (3.10)$$
$$r_4(x, y) = 1 - h_5(y) H_4(x, y), \qquad\qquad (3.11)$$

where again $H_i(x, y) = \int_y^x h_i$.

It is instructive to write the $R$-matrix as

$$R = H_4(x, y) \begin{pmatrix} h_5(x) & 0 & 0 & 0 \\ 0 & 1 & 0 & 0 \\ 0 & 0 & h_5(x) h_5(y) & 0 \\ 0 & 0 & 0 & -h_5(y) \end{pmatrix} + \begin{pmatrix} 1 & 0 & 0 & 0 \\ 0 & 0 & 1 & 0 \\ 0 & 1 & h_5(y) - h_5(x) & 0 \\ 0 & 0 & 0 & 1 \end{pmatrix}. \qquad (3.12)$$

We see that $h_5$ gives rise to the non-difference nature of this solution. In particular, when $h_5$ is constant the $R$-matrix reduces to an $R$-matrix of XXZ type. It is easy to show that it satisfies the Yang-Baxter equation and the correct boundary conditions. This model can be mapped by a twist into the solution A of the pure colored Yang-Baxter equation considered in [60].

**8-vertex A**    In the case $h_6 \neq 0$, the integrability constraint gives that

$$h_4 = h_3 = c_3 h_6, \qquad h_5 = 0, \qquad h_7 = c_7 h_6 e^{4H_2}, \qquad h_8 = c_8 h_6 e^{-4H_2}, \qquad (3.13)$$

where $c_i$ are constants. The resulting Hamiltonian is that of the XYZ spin chain [13,38] under our identifications.

**8-vertex B**    In the case when $h_6 = 0$, we find the following differential equations

$$\frac{\dot{h}_7}{h_7} = 4h_2 + \frac{\dot{h}_3 + \dot{h}_4}{h_3 + h_4} + 4\frac{h_3 - h_4}{h_3 + h_4}h_5, \qquad (3.14)$$

$$\frac{\dot{h}_8}{h_8} = -4h_2 + \frac{\dot{h}_3 + \dot{h}_4}{h_3 + h_4} + 4\frac{h_3 - h_4}{h_3 + h_4}h_5, \qquad (3.15)$$

$$\frac{\dot{h}_5}{h_5} = -\frac{h_3^2 - h_4^2}{4h_5} + \frac{\dot{h}_3 + \dot{h}_4}{h_3 + h_4} + 4\frac{h_3 - h_4}{h_3 + h_4}h_5. \qquad (3.16)$$

We use a local basis transformation to set $h_2 = 0$ and then these equations are solved by

$$h_5 = -\frac{1}{4}(h_3 + h_4)\tanh(H_3 - H_4 + c_5), \qquad (3.17)$$

$$h_7 = c_7 \frac{h_3 + h_4}{\cosh(H_3 - H_4 + c_5)}, \qquad (3.18)$$

$$h_8 = c_8 \frac{h_3 + h_4}{\cosh(H_3 - H_4 + c_5)}. \qquad (3.19)$$

By using a local basis transformation we can set $c_8 = c_7$ and after applying further identifications the remaining functions can be brought to the following form

$$h_3 = \frac{1}{2}\csc(\eta(v))(2 - \dot{\eta}(v)), \qquad (3.20)$$

$$h_4 = \frac{1}{2}\csc(\eta(v))(2 + \dot{\eta}(v)), \qquad (3.21)$$

where $\eta$ is some free function. This further results in $h_7 = h_8 = 2c_7 := k$, which all together imply that $r_5 = r_6 = 1$ and $r_7 = r_8$ for the $R$-matrix. The remaining functions are easily determined from the Sutherland equations and we find

$$r_8(u,v) = k\frac{\mathrm{sn}(u-v,k^2)\mathrm{cn}(u-v,k^2)}{\mathrm{dn}(u-v,k^2)}, \qquad (3.22)$$

where $\mathrm{sn},\mathrm{cn},\mathrm{dn}$ are the usual Jacobi elliptic functions with modulus $k^2$ and

$$r_1 = \frac{1}{\sqrt{\sin\eta(u)}\sqrt{\sin\eta(v)}}\left[\sin\eta_+\frac{\mathrm{cn}}{\mathrm{dn}} - \cos\eta_+\mathrm{sn}\right], \qquad (3.23)$$

$$r_2 = \frac{1}{\sqrt{\sin\eta(u)}\sqrt{\sin\eta(v)}}\left[\cos\eta_-\mathrm{sn} + \sin\eta_-\frac{\mathrm{cn}}{\mathrm{dn}}\right], \qquad (3.24)$$

$$r_3 = \frac{1}{\sqrt{\sin\eta(u)}\sqrt{\sin\eta(v)}}\left[\cos\eta_-\mathrm{sn} - \sin\eta_-\frac{\mathrm{cn}}{\mathrm{dn}}\right], \qquad (3.25)$$

$$r_4 = \frac{1}{\sqrt{\sin\eta(u)}\sqrt{\sin\eta(v)}}\left[\sin\eta_+\frac{\mathrm{cn}}{\mathrm{dn}} + \cos\eta_+\mathrm{sn}\right], \qquad (3.26)$$

where $\eta_\pm = \frac{\eta(u)\pm\eta(v)}{2}$ and all the Jacobi elliptic functions depend on the difference $u-v$, *i.e.* $\mathrm{sn} = \mathrm{sn}(u-v,k^2)$. This solution indeed satisfies the Yang-Baxter equation and has the correct

boundary conditions. Moreover, it is easy to see that in the case where $\eta$ is constant, it becomes of difference form and reduces to the well-known solution found in [38,61,62]. Furthermore, in the limit $k \to \infty$ the $R$-matrix reduces to that of the AdS$_2$ integrable system [20, 41]. We would also like to remark that in [41] we presented the Hamiltonian with a different parameterisation than used here as well as two solutions of the differential equations and hence the $R$-matrix - the two solutions are actually related by a twist thanks to the symmetry $[R_{12}, \sigma_z \otimes \sigma_z] = 0$.

**Off-diagonal model**   As can be seen from (3.14)-(3.16), the cases where $h_5 = 0$ and $h_3 = -h_4$ need special attention due to possible singularities. In particular it is easy to see that by setting $h_5 = 0$ it follows that the Hamiltonian is constant unless $h_3 = -h_4$. And, indeed, in our final expression the limit $h_5 = 0$ corresponds to setting $\eta(x) = \pi/2$.

However, the case $h_3 = -h_4$ warrants special attention. In this case, the entries of the Hamiltonian are

$$h_1 = h_2 = h_5 = h_6 = 0, \qquad h_7 = c_8 \, h_8, \qquad h_3 = -h_4. \tag{3.27}$$

We see that the Hamiltonian for this model only has off-diagonal entries. It can be shown that it is possible to recover this model, starting from the Hamiltonian of 8-vertex B. Since the procedure is highly non-trivial, we explain the steps of this identification.

In order to recover (3.27) we followed the following steps:

1. To the Hamiltonian density $\mathcal{H}_{8VB}$ with entries (3.17)-(3.19), we apply the off-diagonal constant twist

$$U = \begin{pmatrix} 0 & a \\ b & 0 \end{pmatrix} \tag{3.28}$$

   to obtain $\tilde{\mathcal{H}}_{8VB} \to U_1 \mathcal{H}_{8VB} U_1^{-1}$. In order to make $\tilde{\mathcal{H}}_{8VB}$ verify the integrability condition $[\mathbb{Q}_2, \mathbb{Q}_3] = 0$, we fixed one entry of the twist $a \to s_1 \sqrt{c_8} b$, with $s_1 = \pm 1, \pm i$ and we had to impose a constraint on the entries of the Hamiltonian

$$h_3 + h_4 = \alpha_3', \qquad h_3 - h_4 = \frac{\alpha_3' \sinh \alpha_3}{\sqrt{\cosh^2 \alpha_3 + 1}}, \tag{3.29}$$

   with $\alpha_3$ some $\theta$-dependent function. Notice that this twist is non-standard as is does not satisfy (2.14).

2. We apply a diagonal local basis transformation $V(\theta)$. In particular by using (2.8), we first fix $\dot{V}V^{-1}$ to eliminate the elements in the (2,2) and (3,3) positions of the Hamiltonian. Then by solving the differential equations, we fixed the matrix $V(\theta)$.

3. We get an off-diagonal Hamiltonian density and we checked that the sum of the elements at position 2,3 and 3,2 is zero if $s_1$ (defined in step 1) is $\pm i$. Moreover the ratio between elements in 1,4 and 4,1 is constant.

In this way we have recovered model (3.27) from $\mathcal{H}_{8VB}$. Since the twist that we used is non-standard, it is unclear how to easily lift it to the level of the $R$-matrix. Nevertheless, it is easy to solve the Sutherland equations for this model directly and we obtain

$$R_{\text{off-diag}} = \begin{pmatrix} \cosh H_3(u,v) & 0 & 0 & \sin H_7(u,v) \\ 0 & -\sinh H_3(u,v) & \cos H_7(u,v) & 0 \\ 0 & \cos H_7(u,v) & \sinh H_3(u,v) & 0 \\ \sin H_7(u,v) & 0 & 0 & \cosh H_3(u,v) \end{pmatrix}. \tag{3.30}$$

We see that it is of quasi-difference form, meaning all of the dependence on the spectral parameters is of the form $H_3(u) - H_3(v)$ and $H_7(u) - H_7(v)$.

## 3.2 Hermitian solutions

We postpone the classification of all regular $4 \times 4$ solutions of the Yang-Baxter equation to future work due to the complexity of the equations and their solutions. However, there is one interesting physical subcase which we can fully classify. We can classify $4 \times 4$ solutions that give a Hermitian spin chain Hamiltonian. Hence, let's assume we have a Hamiltonian density of the form

$$\mathcal{H} = h^{(r)} + ih^{(i)}, \tag{3.31}$$

where $h^{(r)}$ and $h^{(i)}$ are the real and imaginary parts of the entries of the Hamiltonian. All the functions are now real-valued and imposing hermiticity leaves us with 16 independent *real* functions. Hence in solving the integrability condition we can set both real and imaginary parts to 0. Moreover, we can discard all solutions that have complex numbers in them. This greatly simplifies our computation and, remarkably, we find that all the solutions of this type can be brought into 8 vertex form under our identifications. Note that this does not mean that the corresponding 8 vertex models are Hermitian. There are non-Hermitian 8 vertex models which, after a non-diagonal basis transformations, become Hermitian but are then no longer of 8 vertex type.

# 4 Three-dimensional local Hilbert space

Next we apply our method to $R$-matrices of size $9 \times 9$ corresponding to local Hilbert spaces of dimension 3. In the literature there are many examples of such models including [14, 39, 45–47, 50–52, 63–66].

We consider models whose $R$-matrix and Hamiltonian density commute with the Cartan subalgebra of $\mathfrak{su}(3)$ which are usually referred to as 15-vertex models. These models are a special case of models satisfying the so-called ice rule [50] which states that for an $R$-matrix with components $R^{\mu\alpha}_{\nu\beta}$ in the standard basis we have the constraint

$$R^{\mu\alpha}_{\nu\beta} = 0 \quad \text{unless} \quad \mu + \alpha = \nu + \beta. \tag{4.1}$$

We complete the classification of fifteen-vertex models developed in [39, 64]. As a result of the Cartan symmetry we consider a Hamiltonian density of the form

$$\mathcal{H} = \begin{pmatrix} h_{11} & 0 & 0 & 0 & 0 & 0 & 0 & 0 & 0 \\ 0 & h_{22} & 0 & h_{24} & 0 & 0 & 0 & 0 & 0 \\ 0 & 0 & h_{33} & 0 & 0 & 0 & h_{37} & 0 & 0 \\ 0 & h_{42} & 0 & h_{44} & 0 & 0 & 0 & 0 & 0 \\ 0 & 0 & 0 & 0 & h_{55} & 0 & 0 & 0 & 0 \\ 0 & 0 & 0 & 0 & 0 & h_{66} & 0 & h_{68} & 0 \\ 0 & 0 & h_{73} & 0 & 0 & 0 & h_{77} & 0 & 0 \\ 0 & 0 & 0 & 0 & 0 & h_{86} & 0 & h_{88} & 0 \\ 0 & 0 & 0 & 0 & 0 & 0 & 0 & 0 & h_{99} \end{pmatrix}, \tag{4.2}$$

where $\mathcal{H} := \mathcal{H}(\theta)$ and $h_{ij} := h_{ij}(\theta)$. The corresponding $R$-matrix is of the form

$$R = \begin{pmatrix} r_{11} & 0 & 0 & 0 & 0 & 0 & 0 & 0 & 0 \\ 0 & r_{22} & 0 & r_{24} & 0 & 0 & 0 & 0 & 0 \\ 0 & 0 & r_{33} & 0 & 0 & 0 & r_{37} & 0 & 0 \\ 0 & r_{42} & 0 & r_{44} & 0 & 0 & 0 & 0 & 0 \\ 0 & 0 & 0 & 0 & r_{55} & 0 & 0 & 0 & 0 \\ 0 & 0 & 0 & 0 & 0 & r_{66} & 0 & r_{68} & 0 \\ 0 & 0 & r_{73} & 0 & 0 & 0 & r_{77} & 0 & 0 \\ 0 & 0 & 0 & 0 & 0 & r_{86} & 0 & r_{88} & 0 \\ 0 & 0 & 0 & 0 & 0 & 0 & 0 & 0 & r_{99} \end{pmatrix}, \tag{4.3}$$

where $R := R(u, v)$ and $r_{ij} := r_{ij}(u, v)$.

Applying the procedure described in the previous sections we obtain ten independent models of non-difference form. We had four models for which all $h_{ij}$ in (4.2) and all $r_{ij}$ in (4.3) were nonzero. However, after applying the identifications presented in section 2.2 we learned that these models were actually difference form models disguised by twists, local basis transformations and reparameterizations and corresponded exactly to the models obtained in [39] and [64][2].

The six remaining models are fundamentally of non-difference form, i.e. they cannot be brought to difference form by applying the transformations described in section 2.2, and to our knowledge are new models. An interesting fact is that for the non-difference form $9 \times 9$ cases with Cartan symmetry $\mathfrak{su}(3)$, none of the Hamiltonians are Hermitian. This is in contrast with the $4 \times 4$ case where one can construct models which commute with the Cartan subalgebra of $\mathfrak{su}(2)$ and are still of non-difference form and Hermitian, for example the model (3.1) under a special choice of its free functions.

A curious fact, is that all the cases that are fundamentally of non-difference form for the fifteen vertex models, were the ones coming from singular cases, i.e. cases where we started with some extra zeros in the Hamiltonian (4.2) from the beginning of the procedure.

Below we present the new models of non-difference form. They are divided in two classes depending on whether $p(u, v)$ in equation (4.7) is equal to the permutation operator $P$ (Class 2) or not (Class 1).

## 4.1 Hamiltonian densities

The nonzero matrix elements of the Hamiltonian density (4.2) for each of the six new models are presented in the tables below.

### 4.1.1 Class 1

The matrix elements of the density Hamiltonian for models in Class 1 are given in Table 1

### 4.1.2 Class 2

The matrix elements for the class 2 Hamiltonian densities are given in Table 2
where

$$I(\theta) = -\frac{1}{2} \operatorname{arctanh}\left(e^{2G(\theta)} j(\theta)\right) \tag{4.4}$$

and

$$j(\theta)^2 = e^{-4G(\theta)} + b \tag{4.5}$$

---

[2]In our approach the solutions obtained in [64] are of difference form, in the sense that be mapped to difference form by a twist.

Table 1: Nonzero elements of the Hamiltonian density for models 1-4. Also, $a$, $b$ and $c$ are constants.

| Model | $h_{24}$ | $h_{73}$ | $h_{86}$ | $h_{55}$ | $h_{66}$ | $h_{99}$ |
|:-----:|:--------:|:--------:|:--------:|:--------:|:--------:|:--------:|
| 1 | $b\,e^{-\theta}$ | $a\,e^{\theta}$ | $c$ | 1 | 1 | 1 |
| 2 | $b\,e^{-\theta}$ | $a\,e^{\theta}$ | $c$ | 1 | 1 | 0 |
| 3 | $b\,e^{-\theta}$ | $a\,e^{\theta}$ | $c$ | 0 | 1 | 1 |
| 4 | $b\,e^{-\theta}$ | $a\,e^{\theta}$ | $c$ | 0 | 1 | 0 |

Table 2: Nonzero elements of the Hamiltonian densities for models 5-6. Also, $a$ is a constant.

| Model | $h_{42}$ | $h_{73}$ | $h_{55}$ | $h_{99}$ |
|:-----:|:--------:|:--------:|:--------:|:--------:|
| 5 | $-\frac{2}{3}(g_1 - g_2)e^{2(G_1 - G_2)}$ | $0$ | $2(g_1 - g_2)$ | $2(2g_1 + g_2)$ |
| 6 | $-\frac{2}{3}(g \pm \dot{I})e^{2(G \pm I)}$ | $-\frac{2}{3}a(g \pm \dot{I})e^{2(G \pm I)}$ | $2(g \pm \dot{I})$ | $2(g \mp \dot{I})$ |

and $\dot{I} \equiv \frac{dI(\theta)}{d\theta}$. Also $g_1$, $g_2$, and $g$ are free functions of $\theta$; $a$ and $b$ are constants and $G_i(\theta) = \int^{\theta} g_i(\phi)d\phi$, $i = 1, 2$ while $G(\theta) = \int^{\theta} g(\phi)d\phi$.

## 4.2 *R*-matrices

By exploiting identifications we found that we could bring all of the corresponding *R*-matrices to a form closely resembling the XXX spin chain which we remind the reader is of the form

$$R(u) = uI + P, \tag{4.6}$$

where $I$ denotes the identity operator and $P$ is the permutation operator. We found that we can always write the *R*-matrices for this section in the form

$$R(u, v) = f(u, v)d(u, v) + p(u, v), \tag{4.7}$$

where $f(u, v)$ is some function, $d(u, v)$ is a diagonal $9 \times 9$ matrix and $p(u, v)$ is a matrix with the same entries being non-zero as the usual permutation operator in the standard basis. In fact for models 5 and 6 this operator is exactly the permutation operator.

### 4.2.1 Class 1

The models in Class 1 have in common the fact that they all possess the same $p_i$ given by

$$p = P - (1 - e^{u-v})E_{86} \quad \text{and} \quad f = 2\sinh\left(\frac{u-v}{2}\right), \tag{4.8}$$

where $E_{86}$ is a matrix with 1 in position $(8, 6)$ and 0 everywhere else.

All of the diagonal matrices for these models can be written in the following form

$$d = a\,e^{\frac{u+v}{2}}E_{33} + b\,e^{-\frac{(u+v)}{2}}E_{44} + A\,e^{\frac{u-v}{2}}E_{55} + c\,e^{\frac{u-v}{2}}E_{66} + B\,e^{\frac{u-v}{2}}E_{99}, \tag{4.9}$$

where $E_{kk}$ denotes the matrix with 1 in position $(k, k)$ and 0 everywhere else.

Table 3: Values of the parameters $A$ and $B$.

| Model | $A$ | $B$ |
|:---:|:---:|:---:|
| 1 | 1 | 1 |
| 2 | 1 | 0 |
| 3 | 0 | 1 |
| 4 | 0 | 0 |

### 4.2.2 Class 2

This class has two models and they both have $p$ equal to permutation, *i.e.*,

$$p = P. \tag{4.10}$$

**Model 5**   It is described by

$$f = f^{(5)}(u, v) = 2 \sinh(H_-), \tag{4.11}$$

and

$$d = d_5(u, v) = -\frac{1}{3} e^{H_+} E_{22} + e^{H_-} E_{55} + e^F \frac{\sinh F}{\sinh H_-} E_{99}, \tag{4.12}$$

where $H_\pm = G_1^\pm - G_2^\pm$, $F = 2G_1^- + G_2^-$ and $G_i^\pm = G_i(u) \pm G_i(v)$, $i = 1, 2$.

**Model 6**   It is given by

$$f = f^{(6)}(u, v) = -\frac{2}{3} \sinh(G_- \pm I_-) \tag{4.13}$$

and

$$d = d_6(u, v) = e^{G_+ \pm I_+}(E_{22} + a E_{33}) - 3 e^{G_- \pm I_-} E_{55} - 3 e^{G_- \mp I_-} \frac{\sinh(G_- \mp I_-)}{\sinh(G_- \pm I_-)} E_{99}, \tag{4.14}$$

where $G_\pm = G(u) - G(v)$, $I_\pm = I(u) \pm I(v)$ and $I(u)$ is defined in (4.4) and (4.5).

Notice that model 6 actually defines two separate models due to the choice $\pm$ of signs. They are independent due to the nontrivial dependence of $I(u)$ on $j(u)$ (see (4.4) and (4.5)) and cannot be mapped to each other using any of the transformations in section 2.2.

In order to check the YBE for model 6 in `Mathematica` one needs to be careful with the choice of branch in $j(u)$. The best approach is to substitute $R(u, v)$ in the YBE without specifying $j(u)$, then simplify as most as possible and only then substitute $j(u)^2$ as in equation (4.5). By doing in this way, one never actually needs to choose a branch and the YBE is immediately satisfied. This was already implemented in our `Mathematica` notebook.

## 5   Four-dimensional local Hilbert space

We now apply our method to the case where the local Hilbert space is of dimension 4. In order to have a manageable set-up we restrict ourselves to models which have $\mathfrak{su}(2) \oplus \mathfrak{su}(2)$

symmetry. This class of solutions of the YBE contains important $R$-matrices which correspond to the Hubbard model and AdS/CFT. Moreover, in [40] we also discovered some interesting new models whose $R$-matrix was of difference form. Following it, we see that there are two classes of models with $\mathfrak{su}(2) \oplus \mathfrak{su}(2)$ symmetry. The first class are models where both $\mathfrak{su}(2)$ transform in a four-dimensional representation. These models are of $\mathfrak{so}(4)$ type via the isomorphism $\mathfrak{so}(4) \sim \mathfrak{su}(2) \oplus \mathfrak{su}(2)$. The second class are models where the $\mathfrak{su}(2)$ are represented two-dimensionally. The Hubbard model falls into this category. Finally, we discuss some generalisations of the Hubbard model obtained by taking the Hamiltonian of the free Hubbard model and including the most general possible potential term which preserves fermion number, which allows to interpret the model as electrons moving on a one-dimensional lattice or conduction band. The different form analogue of this setting was discussed in [40].

## 5.1  $\mathfrak{so}(4)$ type models

The most general Hamiltonian density underlying this symmetry takes the form

$$\mathcal{H}(\theta) = h_1(\theta)I + h_2(\theta)P + h_3(\theta)K + h_4(\theta)\epsilon_{ijkl}E_{ik} \otimes E_{jl}, \tag{5.1}$$

where $P$ is the permutation operator, $I$ is the $16 \times 16$ identity matrix and $K = E_{ij} \otimes E_{ij}$ with $(E_{ij})_{\alpha\beta} = \delta_{i,\alpha}\delta_{j,\beta}$. Summation over repeated indices is assumed and $i, j, k, l = 1, ..., 4$.

We found only one possible integrable model of non-difference form with the following Hamiltonian

$$\mathcal{H}(\theta) = h_1(\theta)I + h_2(\theta)(P - K) + h_4(\theta)\,\epsilon_{ijkl}\,E_{ik} \otimes E_{jl}. \tag{5.2}$$

This is the non-difference form model corresponding to the usual $\mathfrak{so}(4)$ spin chain, but here the constant coefficients become functions of the spectral parameter. The $R$-matrix corresponding to (5.2) is given by

$$R = e^{H_1(u,v)}\left[\left(H_2(u,v) - \frac{H_4(u,v)^2}{H_2(u,v)+1}\right)I + P - \frac{H_2(u,v)K - H_4(u,v)\epsilon_{ijkl}E_{ik} \otimes E_{jl}}{H_2(u,v)+1}\right], \tag{5.3}$$

where again $H_i(u,v) = \int_v^u h_i$. Notice that this model is indeed manifestly of non-difference form. One function can be absorbed into a normalization ($H_1$) and one can be used in a reparameterization, so we are left with one additional free function. To be more precise, the $R$-matrix (5.3) is of quasi-difference form, in fact the spectral parameters always appear in $H_i(u,v) = H_i(u) - H_i(v)$.

## 5.2  $\mathfrak{su}(2) \oplus \mathfrak{su}(2)$ symmetry

Next we consider the four-dimensional representation of $\mathfrak{su}(2) \oplus \mathfrak{su}(2)$ in which both $\mathfrak{su}(2)$'s have two-dimensional representation.

**General Hamiltonian and $R$-matrix**  It is straightforward to show that an $\mathfrak{su}(2) \oplus \mathfrak{su}(2)$ invariant Hamiltonian density takes the form

$$\mathcal{H}|\phi_a\phi_b\rangle = h_1|\phi_a\phi_b\rangle + h_2|\phi_b\phi_a\rangle + h_3\epsilon_{ab}\epsilon_{\alpha\beta}|\psi_\alpha\psi_\beta\rangle, \tag{5.4}$$

$$\mathcal{H}|\phi_a\psi_\beta\rangle = h_4|\phi_a\psi_\beta\rangle + h_5|\psi_\beta\phi_a\rangle, \tag{5.5}$$

$$\mathcal{H}|\psi_\alpha\phi_b\rangle = h_6|\psi_\alpha\phi_b\rangle + h_7|\phi_b\psi_\alpha\rangle, \tag{5.6}$$

$$\mathcal{H}|\psi_\alpha\psi_\beta\rangle = h_8|\psi_\alpha\psi_\beta\rangle + h_9|\psi_\beta\psi_\alpha\rangle + h_{10}\epsilon_{ab}\epsilon_{\alpha\beta}|\phi_a\phi_b\rangle. \tag{5.7}$$

Table 4: All non-difference models with $h_3 = 0$. We denote constants by $c, c_1, c_2$, $\theta$ dependent functions by $f, g, h, F$ and $F' = f$. We have omitted the explicit $\theta$ dependence in the latter case.

| $\mathcal{H}$ | $h_1$ | $h_2$ | $h_3$ | $h_4$ | $h_5$ | $h_6$ | $h_7$ | $h_8$ | $h_9$ | $h_{10}$ |
|---|---|---|---|---|---|---|---|---|---|---|
| 1 | $\frac{1}{2(\theta^2-1)}$ | $\frac{1}{2}$ | 0 | $\frac{\theta}{1-\theta^2}$ | $\frac{\pm1}{2}\sqrt{\frac{\theta+1}{\theta-1}}$ | $\frac{\theta}{\theta^2-1}$ | $\frac{\pm1}{2}\sqrt{\frac{\theta-1}{\theta+1}}$ | $\frac{1}{2(1-\theta)^2}$ | $\frac{-1}{2}$ | $c$ |
| 2 | $f$ | $h$ | 0 | $g$ | $\frac{ch}{e^{2F}}$ | $-g$ | $\frac{he^{2F}}{c}$ | $-f$ | $\pm h$ | 0 |
| 3 | $f$ | $\pm h$ | 0 | $g$ | $\frac{ch}{e^{2F}}$ | $-g$ | $\frac{he^{2F}}{c}$ | $h-f$ | 0 | 0 |
| 4 | $(c_1+2)f$ | 0 | 0 | $c_1(f-g)$ | $\frac{c_1(c_1+2)g}{c_2e^{2F}}$ | $(c_1+2)(f-g)$ | $c_2e^{2F}g$ | $c_1f$ | 0 | 0 |
| 5 | $f$ | 0 | 0 | 0 | $g$ | 0 | $h$ | $-f$ | 0 | 0 |
| 6 | $f-h$ | 0 | 0 | $f+h$ | $\frac{2h}{c\,e^{2F}}$ | $h-f$ | $2che^{2F}$ | $h-f$ | $\pm2h$ | 0 |

Here $\phi_{1,2}$ and $\psi_{1,2}$ span the two independent $\mathfrak{su}(2)$ fundamental representations. Explicitly in matrix form, the Hamiltonian density is given by

$$\mathcal{H} = \begin{pmatrix}
h_1+h_2 & 0 & 0 & 0 & 0 & 0 & 0 & 0 & 0 & 0 & 0 & 0 & 0 & 0 & 0 & 0 \\
0 & h_1 & 0 & 0 & h_2 & 0 & 0 & 0 & 0 & 0 & 0 & h_{10} & 0 & 0 & -h_{10} & 0 \\
0 & 0 & h_4 & 0 & 0 & 0 & 0 & 0 & h_7 & 0 & 0 & 0 & 0 & 0 & 0 & 0 \\
0 & 0 & 0 & h_4 & 0 & 0 & 0 & 0 & 0 & 0 & 0 & 0 & h_7 & 0 & 0 & 0 \\
0 & h_2 & 0 & 0 & h_1 & 0 & 0 & 0 & 0 & 0 & 0 & -h_{10} & 0 & 0 & h_{10} & 0 \\
0 & 0 & 0 & 0 & 0 & h_1+h_2 & 0 & 0 & 0 & 0 & 0 & 0 & 0 & 0 & 0 & 0 \\
0 & 0 & 0 & 0 & 0 & 0 & h_4 & 0 & 0 & h_7 & 0 & 0 & 0 & 0 & 0 & 0 \\
0 & 0 & 0 & 0 & 0 & 0 & 0 & h_4 & 0 & 0 & 0 & 0 & 0 & h_7 & 0 & 0 \\
0 & 0 & h_5 & 0 & 0 & 0 & 0 & 0 & h_6 & 0 & 0 & 0 & 0 & 0 & 0 & 0 \\
0 & 0 & 0 & 0 & 0 & 0 & h_5 & 0 & 0 & h_6 & 0 & 0 & 0 & 0 & 0 & 0 \\
0 & 0 & 0 & 0 & 0 & 0 & 0 & 0 & 0 & 0 & h_8+h_9 & 0 & 0 & 0 & 0 & 0 \\
0 & h_3 & 0 & 0 & -h_3 & 0 & 0 & 0 & 0 & 0 & 0 & h_8 & 0 & 0 & h_9 & 0 \\
0 & 0 & 0 & h_5 & 0 & 0 & 0 & 0 & 0 & 0 & 0 & 0 & h_6 & 0 & 0 & 0 \\
0 & 0 & 0 & 0 & 0 & 0 & 0 & h_5 & 0 & 0 & 0 & 0 & 0 & h_6 & 0 & 0 \\
0 & -h_3 & 0 & 0 & h_3 & 0 & 0 & 0 & 0 & 0 & 0 & h_9 & 0 & 0 & h_8 & 0 \\
0 & 0 & 0 & 0 & 0 & 0 & 0 & 0 & 0 & 0 & 0 & 0 & 0 & 0 & 0 & h_8+h_9
\end{pmatrix}, \tag{5.8}$$

where the $h_i$'s are dependent on the spectral parameter $\theta$.

Similarly, for the $R$-matrix, we write

$$R|\phi_a\phi_b\rangle = r_1|\phi_a\phi_b\rangle + r_2|\phi_b\phi_a\rangle + r_3\epsilon_{ab}\epsilon_{\alpha\beta}|\psi_\alpha\psi_\beta\rangle, \tag{5.9}$$

$$R|\phi_a\psi_\beta\rangle = r_4|\phi_a\psi_\beta\rangle + r_5|\psi_\beta\phi_a\rangle, \tag{5.10}$$

$$R|\psi_\alpha\phi_b\rangle = r_6|\psi_\alpha\phi_b\rangle + r_7|\phi_b\psi_\alpha\rangle, \tag{5.11}$$

$$R|\psi_\alpha\psi_\beta\rangle = r_8|\psi_\alpha\psi_\beta\rangle + r_9|\psi_\beta\psi_\alpha\rangle + r_{10}\epsilon_{ab}\epsilon_{\alpha\beta}|\phi_a\phi_b\rangle, \tag{5.12}$$

where $r_i = r_i(u, v)$.

**Integrable Hamiltonians** Upon imposing our integrability constraint we find a total of eight integrable models of non-difference form, in particular six of them have $h_3 = 0$ and are listed in Table 4.

To our best knowledge all of these models are new. They have some interesting properties. These models at most either exhibit electron pair formation or electron pair splitting, not both.

Model 1 can only be made Hermitian if $c = 0$ and the dependence on the spectral parameter drops out, models 2 to 6 are Hermitian if we impose some conditions on the functions, see Table 5. More generally, we can relate the Hamiltonians of the models 1-6 and their Hermitian conjugate by a unitary transformation if we impose the conditions[3] on Table 6.

Model 5 is a quadruple embedding of model 6-vertex B of 3.1. This can be seen by applying a constant local basis transformation[4] to the Hamiltonian of model 6 V B after the redefinition $h_5 \to \frac{1}{2} h_4 h_5$, and by making the identifications

$$h_3 \to g, \qquad\qquad h_4 \to h, \qquad\qquad h_4 h_5 \to -f . \qquad\qquad (5.13)$$

Table 5: Conditions on models 1-6 to make them Hermitian. We put the superscript $(r)$ to identify the real part of the functions and constants.

| Model | Reality conditions |
|-------|--------------------|
| 1 | $\theta = 0, \ c = 0$ |
| 2, 3 | $e^{4F^{(r)}} = |c|^2, \ f, h \in \mathbb{R}$ |
| 4 | $\left( e^{4F^{(r)}} = \frac{c_1^{(r)}(c_1^{(r)}+2)}{|c_2|^2} \ \text{or} \ e^{4F^{(r)}} = \frac{1}{|c_2|^2}, c_1^{(r)} = -1 \right), \ c_1, f, g \in \mathbb{R}$ |
| 5 | $g = h^*, f \in \mathbb{R}$ |
| 6 | $e^{-4F^{(r)}} = |c|^2, f, h \in \mathbb{R}$ |

Table 6: Conditions on the functions and the constants of the full Hamiltonian $\mathcal{H}$ of models 1-6 to make them verify $[\mathcal{H}, \mathcal{H}^\dagger] = 0$. We put the superscripts $(r)$ and $(i)$ to identify the real and the complex part of the functions and constants.

| Model | Unitarity conditions |
|-------|----------------------|
| 1 | $\theta^{(r)} = 0, \ c = 0$ |
| 2, 3 | $e^{4F^{(r)}} = |c|^2$ |
| 4 | $e^{4F^{(r)}} = \frac{c_1^{(i)2}+1}{|c_2|^2}, c_1^{(r)} = -1 \ \text{or} \ e^{4F^{(r)}} = \frac{c_1^{(r)}(c_1^{(r)}+2)}{|c_2|^2}, c_1^{(i)} = 0 \ \text{or} \ c_1 = -1$ |
| 5 | $\forall f, g, h$ |
| 6 | $e^{-4F^{(r)}} = |c|^2$ |

**Model 7** This model is the most general of non-difference form with $h_3 \neq 0$. In fact, we will show that model 8 can be obtained from this one by performing a double limit. In order to solve this model we fixed the normalization of the Hamiltonian such that $h_{10} = 1$[5]. We set

---

[3]To find the conditions on Table 6 we used a chain of length 4.

[4]The entries of this matrix should be $V_{ij} = 1 - \delta_{ij}, i, j = 1, 2$.

[5]We can notice that, if $h_{10}$ cannot be normalized to 1, we should impose $h_{10} = 0$ from the beginning and we found a Hamiltonian $\tilde{\mathcal{H}}$. This model is equivalent to model 1 under the transformation $\left( P \, \tilde{\mathcal{H}} \, P \right)^T$.

$h_4 = h_6 = 0$ by using the identifications described in 2.2, after which we get the following set of coupled differential equations

$$h_1 + h_8 = h_2 + h_9 = 0, \qquad h_8 = \frac{(h_5 + h_7)^2}{4h_9} - h_9, \qquad h_3 = h_5 h_7 - h_9^2, \qquad (5.14)$$

$$\dot{h}_5 = 2h_7 h_9 - \frac{h_5(h_5 + h_7)^2}{2h_9}, \qquad \dot{h}_7 = \frac{h_7(h_5 + h_7)^2}{2h_9} - 2h_5 h_9, \qquad \dot{h}_9 = h_7^2 - h_5^2. \qquad (5.15)$$

Summing the first two equations of (5.15) and taking into account the third one, substituting[6]

$$h_5 = \frac{\sqrt{\xi_1}(\xi_2^2 - 1)}{\sqrt{2}\,\xi_2}, \qquad\qquad h_7 = \frac{\sqrt{\xi_1}(\xi_2^2 + 1)}{\sqrt{2}\,\xi_2}, \qquad (5.16)$$

we find that

$$h_9 = 2\Xi_1, \qquad\qquad \xi_2 = \sigma \frac{\sqrt{\Xi_1}\sqrt{8\Xi_1 + c_1}}{\sqrt{\xi_1}}, \qquad (5.17)$$

where $\Xi_i = \int \xi_i$, $\sigma = \pm 1$ and $c_1$ is a constant. To find $\xi_1$ we made the substitution (5.16) in the differential equation for $h_7$ and we get

$$\Xi_1(8\Xi_1 + c_1)\left(2\dot{\xi}_1 - c_1 \Xi_1(8\Xi_1 + c_1)\right) = \xi_1^2(16\Xi_1 + c_1). \qquad (5.18)$$

For general $c_1$ the equation (5.18) can be solved by performing the substitutions $\Xi_1(u) \to w(u)$, $\xi_1(u) \to \dot{w}(u)$ and $\dot{\xi}_1(u) \to \ddot{w}(u)$ and we find that the corresponding differential equation is solved by elliptic functions

$$\xi_1(u) = \frac{i}{8} c_1^2 \mathrm{cs}(z|m)\mathrm{ds}(z|m)\mathrm{ns}(z|m), \qquad (5.19)$$

where $z = \frac{i}{2} c_1(u + c_2)$ and $m = \frac{8c_3}{c_1^2}$, $c_{2,3}$ are constants. To summarize we then get (5.14) together with

$$h_5 - h_7 = i \frac{\sigma}{2} c_1 \mathrm{ds}(z|m), \qquad\qquad h_5 + h_7 = \frac{\sigma}{2} c_1 \mathrm{nc}(z|m)\left(1 - \mathrm{ns}(z|m)^2\right), \qquad (5.20)$$

$$h_9 = -\frac{1}{4} c_1 \mathrm{ns}(z|m)^2. \qquad (5.21)$$

The Hamiltonian found does not depend on any free functions, so it should be equivalent to the Hamiltonian of AdS/CFT. In order to prove this, we compared the Hamiltonian that we found with the one derived from requiring centrally extended $\mathfrak{su}(2|2)$ symmetry [15]. After using an appropriate normalization and shift, the entries of the Hamiltonian of AdS/CFT are

$$h_1 = -h_8, \qquad h_2 = -h_9, \qquad h_3 = -\frac{1}{\alpha^2}, \qquad h_4 = h_6 = 0, \qquad h_{10} = 1, \qquad (5.22)$$

$$h_5 = \frac{1 - x^{-2}}{\alpha(x^- - x^+)}\sqrt{\frac{x^+}{x^-}}, \quad h_7 = \frac{x^+ \dot{x}^-}{\dot{x}^+ x^-} h_5, \quad h_8 = \frac{(h_5 + h_7)^2}{4h_9} - h_9, \quad h_9 = \frac{1 - x^- x^+}{\alpha(x^- - x^+)}, \quad (5.23)$$

where $\alpha$ is a free constant and $x^+$ and $x^-$ the Zhukovksy variables. $x^\pm$ can be conveniently parametrized using elliptic functions [67] as[7]

$$x^\pm = -\frac{1}{4} i\hbar \left(\mathrm{dn}(\zeta|k) + 1\right)\left(\mathrm{cs}(\zeta|k) \pm i\right), \quad k = \frac{16}{\hbar^2} \qquad (5.24)$$

---

[6]For simplicity we will omit the dependence of $\xi_1, \xi_2, \Xi_1$ and $\Xi_2$ on the spectral parameter.

[7]The parameter $\hbar$ here is related to the parameter $g$ of [67] as $\hbar = \frac{2i}{g}$.

we indeed see that the two Hamiltonian densities are the same under

$$\hbar \to \alpha\, c_1, \quad \alpha^2 \to \frac{2}{c_3} \quad , \quad \zeta \to \frac{i}{2} c_1(c_2 + u), \quad \sigma = 1. \tag{5.25}$$

The other choice of $\sigma = -1$ is not independent, in fact it can be related to the previous one by a twist[8]. So, remarkably, by using our method we are naturally lead to the elliptic parameterization of the AdS/CFT $R$-matrix.

**Model 8**  Model 8 can be obtained as a special limit of Model 7. However, since the limit is somewhat singular, let us spell out this case explicitly. If we solve (5.18) for $c_1 = 0$, we get $\xi_1(u) = c_2 e^{c_3 u}$ and so

$$h_1 = h_4 = h_6 = h_8 = 0, \quad h_2 = -\frac{2c_2 e^{c_3 u}}{c_3}, \quad h_3 = -\frac{c_3{}^2}{16}, \quad h_9 = \frac{2c_2 e^{c_3 u}}{c_3}, \tag{5.27}$$

$$h_5 - h_7 = -\sigma \frac{c_3}{2}, \quad h_5 + h_7 = \sigma \frac{4c_2 e^{c_3 u}}{c_3}, \quad h_{10} = 1. \tag{5.28}$$

It is interesting to notice that the limit $c_1 \to 0$ in the Hamiltonian of model 7 is not well defined because some of the Jacobi functions are divergent in this limit. In order to find the correct results one should take the result for general $c_1$ and then follow the steps below

1. Use the relations that relate the Jacobi functions of modulus $k$ with the ones with modulus $1 - \frac{1}{k}$ like: $\text{ns}(i\,x|k) = -i\sqrt{k}\,\text{cs}\left(x\sqrt{k}|1 - \frac{1}{k}\right)$

2. Expand for small $c_1$

3. Rescale $c_3 \to c_1$

4. Perform a second limit for large $u$

5. Relabel the constants $c_1$ and $c_2$ to obtain (5.27) and (5.28).

**Comparison between difference and non-difference form models**  In order to have a complete classification of the models with $\mathfrak{su}(2)\oplus\mathfrak{su}(2)$ symmetry, we compare the models in Table 1 of [40] with the ones in Table 4 using the allowed identifications, i.e. normalization, shift, rescaling and twists. With this, one can see which non-difference form models constructed here reduce to the difference form given in [40]. By doing this comparison we found the correspondence listed in Table 7. We should mention that to verify that model 3 of difference form can be obtained from model 4 of non-difference form one needs to perform a limit because the solution is found for $c_2 = 0$ (pole for $h_5$) and $c_1 = -2$.

We furthemore notice that models 2, 4 and 7 of non-difference form generate more than one independent difference form models and that models 1 and 3 do not have a difference form version.

Finally, models 9, 10 and 11 of [40] cannot be obtained from any non-difference form version, so if we want a complete classification of $16 \times 16$ matrices with $\mathfrak{su}(2)\oplus\mathfrak{su}(2)$ symmetry we should add those three models.

---

[8]One can easily see that to make the changes $h_5 \to -h_5$ and $h_7 \to -h_7$ in (5.8) we can use the following constant twist

$$V = \begin{pmatrix} 1 & 0 & 0 & 0 \\ 0 & 1 & 0 & 0 \\ 0 & 0 & -1 & 0 \\ 0 & 0 & 0 & 1 \end{pmatrix}, \quad W = \begin{pmatrix} 1 & 0 & 0 & 0 \\ 0 & 1 & 0 & 0 \\ 0 & 0 & 1 & 0 \\ 0 & 0 & 0 & -1 \end{pmatrix}, \quad \mathcal{H}_{\sigma=1} = (V \otimes W)\mathcal{H}_{\sigma=-1}(V \otimes W)^{-1}. \tag{5.26}$$

Table 7: Correspondence between difference form models in Table 1 of [40] and non-difference form of Table 4.

| Difference form | Non-difference form |
|:---:|:---:|
| 1 | 4 |
| 2 | 4, 5 |
| 3 | 4 |
| 4 | 3 ($h_2 = h$) |
| 5 | 2 ($h_9 = h$) |
| 6 | 6 |
| 7 | 2 ($h_9 = -h$) |
| 8 | 7 |
| 12 | 7, 8 |

**Solving Sutherland**  After solving the Sutherland equations we successfully found a unique regular $R$-matrix corresponding to each integrable Hamiltonian. Most of the equations were straightforward to solve. In particular, we can show that from the Sutherland equations for model 5 we can derive the second order version of the Riccati equation as in 6-vertex B of section 3.1.

In the following $c, c_1, c_2$ are constants, $F_\pm = F(u) \pm F(v)$ and similarly for $G$ and $H$, $r_i = r_i(u, v)$ and $\sigma = \pm 1$.

Explicitly, the entries of the $R$-matrices that we got are

**Model 1**

$$r_1 = \frac{-r_{10}\sqrt{1+v}}{2c\sqrt{r_5}\sqrt{1+u}}, \quad r_2 = \frac{-r_1 r_9}{r_8}, \quad r_3 = 0, \qquad r_4 = \pm r_1\sqrt{\frac{u+1}{u-1}}, \quad r_5 = \frac{\sqrt{1-v^2}}{\sqrt{1-u^2}}, \quad (5.29)$$

$$r_6 = \pm r_1\sqrt{\frac{v-1}{v+1}}, \qquad r_7 = \frac{1}{r_5}, \qquad r_8 = -\frac{r_4 r_6}{r_1}, \quad r_9 = \frac{2r_8}{v-u}, \qquad r_{10} = c(v-u); \quad (5.30)$$

**Model 2**

$$r_1 = H_- e^{F_-}, \qquad r_2 = e^{F_-}, \qquad r_3 = 0, \qquad r_4 = cH_- e^{-F_+}, \qquad r_5 = e^{G_-}, \qquad (5.31)$$

$$r_6 = \frac{H_- e^{F_+}}{c}, \qquad r_7 = e^{-G_-}, \qquad r_8 = \pm H_- e^{-F_-}, \qquad r_9 = e^{-F_-}, \qquad r_{10} = 0; \qquad (5.32)$$

**Model 3**

$$r_1 = \pm H_- e^{F_-}, \qquad r_2 = e^{F_-}, \qquad r_3 = 0, \qquad r_4 = \frac{c\,H_-}{e^{F_+}}, \qquad r_5 = e^{G_-}, \qquad (5.33)$$

$$r_6 = \frac{H_- e^{F_+}}{c}, \qquad r_7 = e^{-G_-}, \qquad r_8 = 0, \qquad r_9 = \frac{(H_- + 1)}{e^{F_-}}, \qquad r_{10} = 0; \qquad (5.34)$$

**Model 4**

$$r_1 = r_3 = r_8 = r_{10} = 0, \quad r_2 = \frac{\left((c_1 + 2)e^{2G_-} - c_1\right) r_7}{2}, \quad r_4 = \frac{c_1(c_1 + 2)(e^{2G_-} - 1) r_7}{2c_2 e^{2F(u)}}, \quad (5.35)$$

$$r_5 = e^{c_1(F_- - G_-)}, \qquad r_6 = \frac{c_2^2 e^{2F_+} r_4}{c_1(c_1 + 2)}, \qquad r_7 = e^{(2+c_1)(F_- - G_-)}, \quad (5.36)$$

$$r_9 = e^{-2F_-} r_2; \quad (5.37)$$

**Model 5**

$$r_1 = 0, \qquad r_2 = \frac{H_- f(v)}{h(v)} + 1, \qquad r_3 = 0, \qquad r_4 = \frac{1}{H_-} - \frac{r_2 r_9}{H_-}, \qquad r_5 = 1, \quad (5.38)$$

$$r_6 = H_-, \qquad r_7 = 1, \qquad r_8 = 0, \qquad r_9 = 1 - \frac{H_- f(u)}{h(u)}, \qquad r_{10} = 0. \quad (5.39)$$

It is important to mention that to solve this model we introduced a reparameterization of the spectral parameter, for which

$$u \mapsto x(u) = \int^u \frac{\left(f\dot{h} - h\dot{f}\right)}{h(f^2 - gh)}. \quad (5.40)$$

Only taking this into account the $R$-matrix satisfies the YBE and the boundary conditions.

**Model 6**

$$r_1 = r_3 = r_{10} = 0, \qquad r_2 = e^{F_- + H_-}(1 - 2H_-), \qquad r_4 = \frac{2H_- e^{H_-}}{c\, e^{F_+}}, \qquad r_5 = e^{F_- + H_-}, \quad (5.41)$$

$$r_6 = 2cH_- e^{F_+ + H_-}, \qquad r_7 = \frac{e^{H_-}}{e^{F_-}}, \qquad r_8 = \pm 2H_- \frac{e^{H_-}}{e^{F_-}}, \qquad r_9 = \frac{e^{H_-}}{e^{F_-}}. \quad (5.42)$$

**Model 7**  The $R$-matrix for this model is the AdS/CFT $R$-matrix derived in [15, 68] in the string frame.

**Model 8**

$$r_1 = \frac{e^{-\frac{1}{4}c_3(u+v)} \left(c_3^2 \left(e^{\frac{c_3 u}{2}} - e^{\frac{c_3 v}{2}}\right)^2 - 16c_2 e^{c_3(u+v)} \sinh\left(\frac{1}{2}c_3(u-v)\right)\right)}{2c_3^2 \left(e^{\frac{c_3 u}{2}} + e^{\frac{c_3 v}{2}}\right)}, \quad (5.43)$$

$$r_2 = \frac{1}{\cosh\left(\frac{1}{4}c_3(u-v)\right)}, \quad r_3 = \frac{1}{4}c_3 \tanh\left(\frac{1}{4}c_3(u-v)\right), \quad r_5 = r_7 = \frac{r_2}{r_9} = -\frac{c_3^2 r_{10}}{16 r_3} = 1 \quad (5.44)$$

$$r_4 = -\frac{e^{-\frac{1}{4}c_3(u+v)} \left(e^{\frac{c_3 u}{2}} - e^{\frac{c_3 v}{2}}\right)\left(c_3^2 - 8c_2 e^{\frac{1}{2}c_3(u+v)}\right)}{2c_3^2 \sigma} \quad (5.45)$$

$$r_6 = \frac{8c_2 e^{\frac{1}{4}c_3(u+v)} \left(e^{\frac{c_3 u}{2}} - e^{\frac{c_3 v}{2}}\right)}{c_3^2 \sigma} - r_4, \qquad r_8 = \left(r_4 + r_6\right)\sigma + r_1. \quad (5.46)$$

We gave here the full classification of integrable models with $\mathfrak{su}(2) \oplus \mathfrak{su}(2)$ symmetry. In particular we showed that model 7 corresponds to the $\text{AdS}_5 \times S^5$ integrable system derived in [15, 68]. This model, as shown in [69] contains the one-dimensional Hubbard model. Our next goal is to address the question of finding new Hubbard-type solutions with a more general form.

### 5.3   Generalised Hubbard model

The Hubbard model is an integrable spin chain with 4 dimensional local Hilbert space identified with two bosons and two fermions. The $R$-matrix was constructed by Shastry [55] and is notably of non-difference form. It was known early on that the model possesses $\mathfrak{su}(2) \oplus \mathfrak{su}(2)$ symmetry, but this is not enough to uniquely fix the $R$-matrix. It was later found that the $\mathfrak{su}(2)\oplus\mathfrak{su}(2)$ could be embedded into the centrally extended $\mathfrak{su}(2|2)$ superalgebra which arises naturally with the worldsheet $S$-matrix of the AdS$_5 \times S^5$ integrable system [69] and is equivalent to the Shastry $R$-matrix. Hence, the Hubbard model is of particular significance in the context of the AdS/CFT correspondence and the corresponding integrable structures.

   In this respect we are motivated to look for new integrable models beyond the conventional ones relevant for AdS/CFT integrability potentially associated with new quantum algebras [70,71]. In order to make progress in this direction we implement an ansatz which preserves fermion number but can violate spin conservation as was done in [40]

$$\mathcal{H} = \mathcal{H}_{\text{Kin}} + K_{\text{Flip}} + K_{\text{Pair}} + V, \tag{5.47}$$

where $\mathcal{H}_{\text{Kin}}$ is the kinetic term of the free Hubbard model and $K_{\text{Pair}}$ and $K_{\text{Flip}}$ are further kinetic terms which describe the hopping of a pair of electrons and a term which flips the spins of the electrons on neighbouring sites, respectively, and $V$ is a general potential term, and the precise form of these operators can be found in [40]. The total space of solutions is very large, but we can single out solutions that have the maximal amount of non-zero entries. In particular, demanding that all entries are non-zero, we only find one independent solution. Remarkably it reduces to an $R$-matrix of difference form. The Hamiltonian is given by

$$\mathcal{H} = \begin{pmatrix}
-\lambda & 0 & 0 & 0 & 0 & 0 & 0 & 0 & 0 & 0 & 0 & 0 & 0 & 0 & 0 & 0 \\
0 & \lambda & 0 & 0 & 0 & 0 & 0 & 0 & 0 & 0 & 0 & \rho_2 & 0 & 0 & -\rho_2 & 0 \\
0 & 0 & 0 & 0 & 0 & 0 & 0 & 0 & \rho_1 & 0 & 0 & 0 & 0 & 0 & 0 & 0 \\
0 & 0 & 0 & 0 & 0 & 0 & 0 & 0 & 0 & 0 & 0 & 0 & \rho_1 & 0 & 0 & 0 \\
0 & 0 & 0 & 0 & \lambda & 0 & 0 & 0 & 0 & 0 & 0 & -\rho_2 & 0 & 0 & \rho_2 & 0 \\
0 & 0 & 0 & 0 & 0 & -\lambda & 0 & 0 & 0 & 0 & 0 & 0 & 0 & 0 & 0 & 0 \\
0 & 0 & 0 & 0 & 0 & 0 & 0 & 0 & 0 & \rho_1 & 0 & 0 & 0 & 0 & 0 & 0 \\
0 & 0 & 0 & 0 & 0 & 0 & 0 & 0 & 0 & 0 & 0 & 0 & 0 & \rho_1 & 0 & 0 \\
0 & 0 & -\rho_1 & 0 & 0 & 0 & 0 & 0 & 0 & 0 & 0 & 0 & 0 & 0 & 0 & 0 \\
0 & 0 & 0 & 0 & 0 & 0 & -\rho_1 & 0 & 0 & 0 & 0 & 0 & 0 & 0 & 0 & 0 \\
0 & 0 & 0 & 0 & 0 & 0 & 0 & 0 & 0 & 0 & 0 & 0 & 0 & 0 & \tau\lambda & \\
0 & -\xi & 0 & 0 & \xi & 0 & 0 & 0 & 0 & 0 & 0 & 0 & 0 & 0 & -\lambda & 0 \\
0 & 0 & 0 & -\rho_1 & 0 & 0 & 0 & 0 & 0 & 0 & 0 & 0 & 0 & 0 & 0 & 0 \\
0 & 0 & 0 & 0 & 0 & 0 & 0 & -\rho_1 & 0 & 0 & 0 & 0 & 0 & 0 & 0 & 0 \\
0 & \xi & 0 & 0 & -\xi & 0 & 0 & 0 & 0 & 0 & -\lambda & 0 & 0 & 0 & 0 & 0 \\
0 & 0 & 0 & 0 & 0 & 0 & 0 & 0 & 0 & 0 & \frac{\lambda}{\tau} & 0 & 0 & 0 & 0 & 0
\end{pmatrix}, \tag{5.48}$$

where

$$\rho_1 = i\sqrt{\lambda^2 - 1}, \quad \text{and} \quad \rho_2 = \frac{1 - \lambda^2}{\xi} \tag{5.49}$$

and $\tau$, $\lambda$ and $\xi$ are constant parameters.

The corresponding R-matrix is given by

$$R = \begin{pmatrix}
r_1 & 0 & 0 & 0 & 0 & 0 & 0 & 0 & 0 & 0 & 0 & 0 & 0 & 0 & 0 & 0 \\
0 & r_2 & 0 & 0 & r_{11} & 0 & 0 & 0 & 0 & 0 & 0 & -r_8 & 0 & 0 & r_8 & 0 \\
0 & 0 & r_4 & 0 & 0 & 0 & 0 & 0 & r_{10} & 0 & 0 & 0 & 0 & 0 & 0 & 0 \\
0 & 0 & 0 & r_4 & 0 & 0 & 0 & 0 & 0 & 0 & 0 & 0 & r_{10} & 0 & 0 & 0 \\
0 & r_{11} & 0 & 0 & r_2 & 0 & 0 & 0 & 0 & 0 & 0 & r_8 & 0 & 0 & -r_8 & 0 \\
0 & 0 & 0 & 0 & 0 & r_1 & 0 & 0 & 0 & 0 & 0 & 0 & 0 & 0 & 0 & 0 \\
0 & 0 & 0 & 0 & 0 & 0 & r_4 & 0 & 0 & r_{10} & 0 & 0 & 0 & 0 & 0 & 0 \\
0 & 0 & 0 & 0 & 0 & 0 & 0 & r_4 & 0 & 0 & 0 & 0 & 0 & r_{10} & 0 & 0 \\
0 & 0 & r_7 & 0 & 0 & 0 & 0 & 0 & r_3 & 0 & 0 & 0 & 0 & 0 & 0 & 0 \\
0 & 0 & 0 & 0 & 0 & 0 & r_7 & 0 & 0 & r_3 & 0 & 0 & 0 & 0 & 0 & 0 \\
0 & 0 & 0 & 0 & 0 & 0 & 0 & 0 & 0 & 0 & r_5 & 0 & 0 & 0 & 0 & r_{13} \\
0 & -r_9 & 0 & 0 & r_9 & 0 & 0 & 0 & 0 & 0 & 0 & r_6 & 0 & 0 & r_{12} & 0 \\
0 & 0 & 0 & r_7 & 0 & 0 & 0 & 0 & 0 & 0 & 0 & 0 & r_3 & 0 & 0 & 0 \\
0 & 0 & 0 & 0 & 0 & 0 & 0 & r_7 & 0 & 0 & 0 & 0 & 0 & r_3 & 0 & 0 \\
0 & r_9 & 0 & 0 & -r_9 & 0 & 0 & 0 & 0 & 0 & 0 & r_{12} & 0 & 0 & r_6 & 0 \\
0 & 0 & 0 & 0 & 0 & 0 & 0 & 0 & 0 & 0 & r_{14} & 0 & 0 & 0 & 0 & r_5
\end{pmatrix}, \tag{5.50}$$

where

$$r_1 = \cosh u - \lambda \sinh u, \qquad r_2 = \frac{(1-\lambda^2)\sinh u \tanh u}{1 - \lambda \tanh u},$$

$$r_3 = i\sqrt{\lambda^2 - 1}\sinh u = -r_4, \qquad r_5 = \cosh u,$$

$$r_6 = -\frac{\sinh u\,(\lambda - \tanh u)}{1 - \lambda \tanh u}, \qquad r_7 = 1 = r_{10},$$

$$r_8 = \frac{(1 - \lambda^2 \tanh u)}{\xi\,(1 - \lambda \tanh u)}, \qquad r_9 = -\frac{\xi \tanh u}{1 - \lambda \tanh u},$$

$$r_{11} = \frac{\operatorname{sech} u}{1 - \lambda \tanh u}, \qquad r_{12} = \frac{\operatorname{sech} u\,(2 - \lambda \sinh 2u + 2\lambda^2 \sinh^2 u)}{2\,(1 - \lambda \tanh u)},$$

$$r_{13} = \tau\,\lambda \sinh u, \qquad r_{14} = \frac{\lambda \sinh u}{\tau}. \tag{5.51}$$

There will be new models with lower number of non-zero parameters, but as mentioned before, the solution space is very large and the full classification remains an open and interesting question.

# 6 Discussion and conclusions

In this paper we have classified various types of integrable systems and found a plethora of new models generalizing a method based on the boost operator initially put forward in [1]. By starting with a generic Hamiltonian we constrained it to potentially belong to an integrable model by imposing that it commutes with the first higher conserved charge generated by the boost operator. In all cases we showed that this condition is sufficient and we were able to subsequently derive the corresponding $R$-matrices, guaranteeing integrability. For $4 \times 4$ models we reviewed the classification of 8-and-lower vertex models originally presented in [41]. We also proved that any Hermitian integrable Hamiltonian can be reconducted (using a local basis transformation) to be 8 V type. Next, we examined $9 \times 9$ models and completely classified all 15-vertex models satisfying the ice-rule. Finally, for $16 \times 16$ $R$-matrices we classified all models with $\mathfrak{su}(2) \oplus \mathfrak{su}(2)$ symmetry.

There are various interesting avenues for future research. We discuss some of them here. One natural direction involves applications to holography and the integrable systems which appear in that context. Generalised Shastry-type models provide a base for a search of new types of solutions that are relevant for $AdS_{4,5}$ integrable models. In particular it would be interesting

to search for new deformations of the $AdS_{4,5}$ S-matrix and establish potential contact with $q$-deformations of the underlying twisted Hopf algebra as for $\eta$-deformed $AdS_5 \times S^5$ [72,73] or for $\lambda$-deformed systems which appear to be non-ultralocal [74]. We already discussed in [41] that the $AdS_2$ S-matrix could be embedded into the $4 \times 4$ model 8VB and admitted a one-parameter deformation. Similarly we showed that the S-matrices of $AdS_3$ governing the scattering of particles with the same chirality could be embedded into both 6VB and 8VB, and both had tunable parameters which correspond to sources of deformations. It would be highly interesting to find a physical interpretation for these parameters and to determine the symmetry algebras of the resulting S-matrices and, in the case of $AdS_3$, to check if the deformations of the same chirality S-matrices induce a corresponding deformation of the opposite chirality S-matrices. As well as this, in this paper we demonstrated that there are no integrable deformations of the $AdS_5/CFT_4$ S-matrix compatible with $\mathfrak{su}(2) \oplus \mathfrak{su}(2)$ symmetry. However, integrable deformations which do not have this symmetry do exist, for example the $q$-deformed model [72] and associated quantum algebra [71,75], and it may still be possible to construct integrable deformations which do not have this symmetry by using a perturbative approach. For example, one can start from a given integrable Hamiltonian $\mathcal{H}_{12}^{(0)}$ and deform it to $\mathcal{H}_{12}^{0} + \epsilon \, \mathcal{H}_{12}^{(1)}$. By imposing the commutativity of the first two conserved charges, one will obtain a set of linear equations for the coefficients of $\mathcal{H}_{12}^{(1)}$. After factoring out trivial deformations arising from identifications, the resulting set of equations should be quite tractable.

It would be also very interesting to start from Hamiltonians with more vertices and less symmetry. For a Hilbert space of dimension three for example, one could consider 19-vertex Hamiltonians and do a full classification of these models. It is likely that many new non-difference form models could be found. It would be interesting to verify if some of the models we found in the 15-vertex case are actually special cases of more general 19-vertex models.

Furthermore, for Hilbert spaces of dimension 4 we obtained a correspondence between some the non-difference form models found and the difference one of [40]. In particular we found that models 9, 10 and 11 of [40] cannot be obtained from any of the non-difference form models presented in this paper. It would be very interesting to check if these models can be obtained as special limits from other non-difference form models with less symmetry then the $\mathfrak{su}(2) \oplus \mathfrak{su}(2)$ considered here.

Another question that could be addressed is the construction of finite length open spin chains for all the new models constructed in this paper. In order to do that, the first step would be the construction of all possible integrable boundary conditions, meaning all solutions of the Boundary Yang-Baxter equation [76,77] for each of the R-matrices introduced here.

Remarkably, all of our solutions of the Yang-Baxter equation can be characterized by the integrability condition $[\mathbb{Q}_2, \mathbb{Q}_3] = 0$. It is unclear to us why this is the case. Indeed, all the reverse lines in the flowchart, Figure 1, can be shown to hold. The reverse arrows that we exploit here, however, appear to be valid as well and it seems to indicate an equivalence relation. It would be very important to understand and prove these relations.

There are also interesting related mathematical questions to be asked. In the case of difference form models the condition $[\mathbb{Q}_2, \mathbb{Q}_3] = 0$ results in a set of cubic polynomial equations for the Hamiltonian entries which seems to be fully equivalent to the Yang-Baxter equation. It would be highly interesting to construct a proof of this claim and in doing so perhaps obtain a closed form expression for the $R$-matrix in terms of the Hamiltonian entries. In this paper we have relied on a brute force approach to solving the constraint $[\mathbb{Q}_2, \mathbb{Q}_3] = 0$ and to a large extent have exhausted the cases where such an approach is applicable.

In order to make more progress it could be important to make use of the extensive toolbox of algebraic geometry. Indeed, $[\mathbb{Q}_2, \mathbb{Q}_3] = 0$ describes an algebraic variety in projective space described by a set of coupled, cubic polynomials. For instance, in the $4 \times 4$ case the integrable models will correspond to algebraic varieties in $\mathbb{CP}^{16}$. It would be very interesting to exactly

understand what the algebraic varieties are that describe integrable models and how exactly they can be characterized.

# Acknowledgements

We would like to thank V. Korepin, V. Kazakov, D. Gurevich, K. Zarembo, L. Takhtadzhan, R. Pimenta and A. Torrielli for discussions. MdL was supported by SFI, the Royal Society and the EPSRC for funding under grants UF160578, RGF\R1\181011, RGF\EA\180167 and 18/EPSRC/3590. C.P. is supported by the grant RGF\R1\181011. A.P. is supported by the grant RGF\EA\180167. A.L.R. is supported by the grant 18/EPSRC/3590. P.R. is supported in part by a Nordita Visiting PhD Fellowship and by SFI and the Royal Society grant UF160578.

# A Non-difference form boost operator and *R*-matrix

In this section we review the construction of the boost operator for non-difference form models. Our exposition closely follows that of [43, 44].

Our starting point is the Sutherland equation

$$[R_{13}R_{12}, \mathcal{H}_{23}(\theta)] = R_{13}R'_{12} - R'_{13}R_{12}, \tag{A.1}$$

where again we denote $R_{ij} := R_{ij}(u, \theta)$ and we remind the reader that $R'$ denotes the derivative with respect to the second argument. We now make the replacement $1 \mapsto a, 2 \mapsto k, 3 \mapsto k+1$, obtaining

$$\left[R_{a,k+1}R_{ak}, \mathcal{H}_{k,k+1}(\theta)\right] = R_{a,k+1}R'_{ak} - R'_{a,k+1}R_{ak}. \tag{A.2}$$

We now consider an infinite spin chain with monodromy matrix $T_a(u, \theta)$ given by

$$T_a(u, \theta) = \dots R_{a1}R_{a0}R_{a,-1} \dots . \tag{A.3}$$

Now take (A.2) and multiply from the left with the product of *R*-matrices $\dots R_{a,k+2}$ and from the right with $R_{a,k-1} \dots$. We then multiply the resulting equation by $k$ and sum over $k$ from $-\infty$ to $\infty$. The two terms on the right hand side of (A.2) telescopically cancel and we are left with

$$\sum_{k=-\infty}^{\infty} k\left[T_a(u, \theta), \mathcal{H}_{k,k+1}(\theta)\right] = \frac{dT_a(u, \theta)}{d\theta}, \tag{A.4}$$

which gives

$$\sum_{k=-\infty}^{\infty} k\left[t(u, \theta), \mathcal{H}_{k,k+1}(\theta)\right] = \frac{dt(u, \theta)}{d\theta} \tag{A.5}$$

after tracing over the auxiliary space. Finally, using the expansion

$$\log t(u, \theta) = \mathbb{Q}_1(\theta) + (u - \theta)\mathbb{Q}_2(\theta) + \frac{1}{2}(u - \theta)^2 \mathbb{Q}_3(\theta) + \dots \tag{A.6}$$

we obtain

$$\mathbb{Q}_{r+1}(\theta) = \sum_{k=-\infty}^{\infty} k[H_{k,k+1}(\theta), \mathbb{Q}_r(\theta)] + \partial_\theta \mathbb{Q}_r(\theta), \quad r = 2, 3, \dots . \tag{A.7}$$

# B Notebook

We have provided a `Mathematica` notebook with the arxiv submission of this paper. The notebook provides a database of all *R*-matrices presented in this publication along with those presented in [1, 40, 41].

A model is defined by three parameters, which we denote in `Mathematica` as `spec`, `dimHS` and `model`. `spec` is a list containing the set of spectral parameters of the *R*-matrix. For a non-difference form *R*-matrix $R(u, v)$ we have `spec = {u,v}` and for a difference-form *R*-matrix $R(u)$ we have `spec = {u}`.

The parameter `dimHS` can take the values $2, 3, 4$ and specifies the dimension of the local spin chain Hilbert space. If `dimHS = n` then the corresponding *R*-matrix is of size $n^2 \times n^2$.

Finally, `model` can take the values $0, 1, 2, \dots$ and specifies, together with `dimHS` and `spec` which of the models in the papers [1, 40, 41] and in this publication we are referring to. The precise map between the value of `model` and *R*-matrices is specified in the notebook.

**R-matrix** The *R*-matrix corresponding to a given model as explained above is obtained by running the command

$$\texttt{rmat[spec, dimHS, model]}$$

For example, in order to obtain the *R*-matrix of 8-vertex B model of section 3.1 in the present paper we run the command

$$\texttt{rmat[\{u,v\}, 2, 3]}$$

**Hamiltonian** The Hamiltonian is obtained in a way similar to the *R*-matrix. We run the command

$$\texttt{hamil[spec, dimHS, model]}$$

where `spec={u}` for non-difference form and `spec={}` for difference form.

**Yang-Baxter equation** To test the Yang-Baxter equation we run the command

$$\texttt{ybe[spec, dimHS, model]}$$

where `spec = {u,v,w}` for a non-difference form model and `spec = {u,v}` for difference form. This command evaluates $R_{12}(u,v)R_{13}(u,w)R_{23}(v,w) - R_{23}(v,w)R_{13}(u,w)R_{12}(u,v)$ for non-difference form and $R_{12}(u-v)R_{13}(u)R_{23}(v) - R_{23}(v)R_{13}(u)R_{12}(u-v)$ for difference form. If the Yang-Baxter equation is satisfied the output is {0}.

**Regularity** Regularity is the condition that

$$R_{12}(0) - \alpha P_{12} = 0, \quad R_{12}(u,u) - \alpha(u)P_{12} = 0, \tag{B.1}$$

where we refer to $\alpha$, $\alpha(u)$ as the regularity coefficient. This is represented in `Mathematica` as `coeffregul[spec, dimHS, model]`. The command

`regularity[spec, dimHS, model]`

computes the l.h.s. of (B.1) which produces {0} if regularity is satisfied.

**Braiding unitarity**   We can also check braiding unitarity which is satisfied if there exists a scalar function $\beta(u)$ or $\beta(u,v)$ such that

$$R_{12}(u)R_{21}(-u) - \beta(u) = 0, \quad R_{12}(u,v)R_{21}(v,u) - \beta(u,v) = 0. \tag{B.2}$$

We refer to $\beta(u), \beta(u,v)$ as the braiding coefficient and represent it in our notebook as `coeffbraid[spec, dimHS, model]`. The command

    braiding[spec, dimHS, model]

computes the l.h.s. of (B.2), which produces {0} if braiding unitarity is satisfied.

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
