# Peer review of "Yang-Baxter and the Boost: splitting the difference"

_SciPost Physics, doi:SciPost Phys. 11, 069 (2021)_

## Round 1 · Referee Report · Anonymous (Referee 2) · 2021-8-31

Report
In this well written paper the authors continue the classification of regular solutions of the Yang-Baxter equation using a boost operator method. The paper is well organised, explains the method clearly and reproduces known solutions, including elliptic ones. They authors also seem to obtain some new solutions that are not of the usual difference form.
I would be happy for the paper to be published as is but would ask the authors to consider the following old problem. I would be very interested in their thoughts. The square-triangle tiling transfer matrix was diagonalised by Widom in the 90s using the coordinate Bethe ansatz, and in https://arxiv.org/abs/solv-int/9611005 it was shown that the model arises as a singular limit from the sl(3) integrable model, i.e. the resulting R-matrix is singular. Invertability can be restored in this case by extending the model and this was used in https://arxiv.org/abs/0809.2392 and later work by the same author to solve a number of combinatorial and geometric problems.
Two other tilings have been shown to be solvable by (nested) coordinate Bethe ansatz: the octagonal rectangle-triangle tiling (https://arxiv.org/abs/solv-int/9602002, https://arxiv.org/abs/solv-int/9610009) and the decagonal rectangle-triangle tiling (https://arxiv.org/abs/cond-mat/9709338), but related R-matrices with a spectral parameter were never found for these models. Would the boost method employed in the paper be able to generate such an R-matrix?
I would be happy for the paper to be published as is but would ask the authors to consider the following old problem. I would be very interested in their thoughts. The square-triangle tiling transfer matrix was diagonalised by Widom in the 90s using the coordinate Bethe ansatz, and in https://arxiv.org/abs/solv-int/9611005 it was shown that the model arises as a singular limit from the sl(3) integrable model, i.e. the resulting R-matrix is singular. Invertability can be restored in this case by extending the model and this was used in https://arxiv.org/abs/0809.2392 and later work by the same author to solve a number of combinatorial and geometric problems.
Two other tilings have been shown to be solvable by (nested) coordinate Bethe ansatz: the octagonal rectangle-triangle tiling (https://arxiv.org/abs/solv-int/9602002, https://arxiv.org/abs/solv-int/9610009) and the decagonal rectangle-triangle tiling (https://arxiv.org/abs/cond-mat/9709338), but related R-matrices with a spectral parameter were never found for these models. Would the boost method employed in the paper be able to generate such an R-matrix?

---

## Round 1 · Referee Report · Anonymous (Referee 1) · 2021-8-31

Report
In this paper, the authors revisit the problem of classify the solution of the Yang–Baxter
equation, further developing a method to obtain solutions of the Yang–Baxter equation, devel-
oped in previous publication by the same authors.
The method starts with the hamiltonian density and uses the boost automorphism in order
to generate the higher commuting charges. Then the hamiltonian is required to commute with
the third charge: this puts necessary conditions on the hamiltonian density for the model to be
integrable. In all the cases treated by the authors in the paper, these necessary conditions are
also sufficient to solve the Sutherland equations giving the R–matrix. This general procedure is
carefully explained in Section 2.1.
Using this procedure the authors treat problems with local Hilbert space of dimension 2, 3
and 4. For the two–dimensional case the authors are able to classify all the models with 8 and
lower vertexes. For the three-dimensional case they classify all R-matrices commuting with the
local Cartan subalgebra of su(3), while for the four–dimensional case they classify all R–matrices
commuting with the su(2) ⊕ su(2).
In conclusion. The paper is carefully written. It presents new and in my opinion interesting
results, that as far as I’ve been able to check are correct and deserve publication. I recommend
the paper for publication.
equation, further developing a method to obtain solutions of the Yang–Baxter equation, devel-
oped in previous publication by the same authors.
The method starts with the hamiltonian density and uses the boost automorphism in order
to generate the higher commuting charges. Then the hamiltonian is required to commute with
the third charge: this puts necessary conditions on the hamiltonian density for the model to be
integrable. In all the cases treated by the authors in the paper, these necessary conditions are
also sufficient to solve the Sutherland equations giving the R–matrix. This general procedure is
carefully explained in Section 2.1.
Using this procedure the authors treat problems with local Hilbert space of dimension 2, 3
and 4. For the two–dimensional case the authors are able to classify all the models with 8 and
lower vertexes. For the three-dimensional case they classify all R-matrices commuting with the
local Cartan subalgebra of su(3), while for the four–dimensional case they classify all R–matrices
commuting with the su(2) ⊕ su(2).
In conclusion. The paper is carefully written. It presents new and in my opinion interesting
results, that as far as I’ve been able to check are correct and deserve publication. I recommend
the paper for publication.

---

## Editorial Decision

published